# *Odoribacter splanchnicus* rescues aging-related intestinal P-glycoprotein damage via GDP-L-fucose secretion

Cheng Cui[1,2,10], Lu Fang[3,4,5,10], Lei Li[4,6,7,10], Xuan Lai[8], Ruitao Zhang[4,6,7], Qi Zhang[3,4], Rong Miao[3,4], Gaofei Hu[3,4], Miao Zhang[1,2], Jie En Valerie Sia[1,2], Jingcheng Chen[1,2], Haodi Chai[1,2], Xinyi Wu[1,2], Zijin Lin[1,2], Fan Zhang[8], Haiyan Li[4,6,7], Lemin Zheng ![ORCID][3,4,9] ✉ & Dongyang Liu ![ORCID][1,2,4,6] ✉

Intestinal P-glycoprotein (P-gp/*ABCB1*) is a key barrier limiting xenobiotic absorption, yet its functional decline with aging is poorly understood. Here, we show that gut microbiota dysbiosis contributes to age-associated P-gp deficiency. Integrated multi-omics analyses of human cohorts and murine models identify *Odoribacter splanchnicus* (*O. splanchnicus*) as a key commensal species whose depletion impairs intestinal P-gp function. Mechanistically, *O. splanchnicus* encodes GDP-mannose 4, 6-dehydratase (GMDS) and GDP-L-fucose synthase (TSTA3), enabling microbial biosynthesis of GDP-L-fucose. This metabolite directly promotes phosphorylation of the eukaryotic translation initiation factor 4E (eIF4E) and activates c-Jun-driven *ABCB1* expression, thereby restoring xenobiotic efflux. These findings establish a microbiota-metabolite-transporter signaling axis that maintains intestinal detoxification, suggesting that targeting either microbes or metabolites could help prevent adverse drug reactions in older adults.

Physiological aging is accompanied by a progressive decline in xenobiotic clearance pathways, including impaired hepatic metabolism and renal excretion, which contribute to altered pharmacokinetics and heightened susceptibility to adverse drug reactions (ADRs) in older adults[1]. Among the various age-related changes, intestinal P-glycoprotein (P-gp/*ABCB1*), an ATP-driven efflux transporter located at the apical membrane of enterocytes, plays a particularly important role in regulating drug absorption and bioavailability[2–4]. Reduced P-gp activity increases systemic exposure of substrate drugs, thereby elevating the risk of clinically significant adverse events[5–7]. Thus, while not the sole determinant, P-gp impairment represents a distinct and potentially modifiable mechanism contributing to age-related ADRs.

This is exemplified by direct oral anticoagulants (DOACs), where reduced intestinal P-gp activity increases bioavailability and plasma concentrations, correlating with higher rates of major bleeding[6]. Clinically, patients aged ≥ 75 years exhibit a 1.4-fold higher incidence of

[1]Drug Clinical Trial Center, Department of Pharmacy, Peking University Third Hospital, Beijing, China. [2]Center of Clinical Medical Research, Institute of Medical Innovation and Research, Peking University Third Hospital, Beijing, China. [3]The Institute of Cardiovascular Sciences, the Institute of Systems Biomedicine, School of Basic Medical Sciences, Health Science Center, Peking University, Beijing, China. [4]State Key Laboratory of Vascular Homeostasis and Remodeling, NHC Key Laboratory of Cardiovascular Molecular Biology and Regulatory Peptides, Peking University, Beijing, China. [5]Research Center for Cardiopulmonary Rehabilitation, University of Health and Rehabilitation Sciences Qingdao Hospital (Qingdao Municipal Hospital), School of Health and Life Sciences, University of Health and Rehabilitation Sciences, Qingdao, China. [6]Beijing Key Laboratory of Cardiovascular Receptors Research, Peking University Third Hospital, Beijing, China. [7]Department of Cardiology and Institute of Vascular Medicine, Peking University Third Hospital, Beijing, China. [8]Geriatrics Department, Peking University Third Hospital, Beijing, China. [9]Beijing Tiantan Hospital, China National Clinical Research Center for Neurological Diseases, Advanced Innovation Center for Human Brain Protection, The Capital Medical University, Beijing, China. [10]These authors contributed equally: Cheng Cui, Lu Fang, Lei Li. ✉e-mail: zhengl@bjmu.edu.cn; liudongyang@vip.sina.com

major bleeding compared to younger individuals, rising to 2.0-fold for intracranial hemorrhage in octogenarians[7,8]. These findings demonstrate how impaired P-gp function critically shapes drug toxicity, even while acting in concert with other aging-related factors, such as hepatic and renal dysfunction[9]. Similarly, reduced P-gp activity elevates systemic exposure to digoxin, contributing to excess neurocardiac adverse events[10–12]. Together, these observations position impaired intestinal P-gp function as a pivotal mechanism linking altered drug exposure to ADRs in older adults, highlighting opportunities for targeted interventions to improve pharmacotherapy safety in geriatric populations.

Emerging evidence implicates gut microbiota dysbiosis as a critical regulator of host-drug interactions, influencing both metabolic enzymes and transporter expression[13–15]. However, most studies remain descriptive, cataloging microbial taxa correlated with pharmacokinetic variability, without identifying causative species, effector metabolites, or molecular pathways that mechanistically link microbiota to transporter regulation[16,17]. This gap is especially relevant in aging, where cumulative exposures and immunosenescence reshape the gut microbiota in parallel with declining P-gp function. We therefore hypothesized that age-related microbial shifts disrupt metabolite-mediated signaling required for intestinal P-gp homeostasis.

To address this, we combined multi-omics profiling of age-stratified human cohorts with mechanistic validation in antibiotic-pretreated mouse models. This strategy was designed to identify specific microbial taxa and metabolites that regulate intestinal P-gp, and to define the molecular pathways underlying their effects.

## Results

### Age-related decline in intestinal P-gp functionality

A pooled analysis of clinical pharmacokinetic studies of FDA-approved P-gp substrates revealed that older adults exhibited approximately 2-fold higher systemic drug exposure and 150% greater inter-individual variability than younger adults (Fig. 1A, B). To isolate the specific effect of aging from potential confounding factors, we analyzed dabigatran etexilate, a canonical intestinal P-gp probe substrate, using retrospective population pharmacokinetic (PopPK) and physiologically-based pharmacokinetic (PBPK) modeling. These analyses demonstrated a markedly reduced intestinal P-gp activity in patients ≥75 years vs those aged 60–74 years, independent of demographics, cardiac output, or renal function[18]. Sensitivity simulations indicated that impaired jejunal and ileal P-gp alone could double systemic exposure, with older adults more frequently exceeding pharmacological thresholds associated with bleeding risk (Fig. 1C, D)[19].

Clinically, pooled randomized controlled trials (RCTs) of P-gp substrate DOACs, including dabigatran etexilate, edoxaban, rivaroxaban, and apixaban, confirmed that patients ≥75 years had a higher risk of major bleeding than those <75 years (RR = 0.53; 95% CI: 0.50–0.57; Figs. S1 and S2), and co-administration of P-gp inhibitors (dabigatran etexilate and edoxaban) further increased bleeding risk (RR = 0.81; 95% CI: 0.72–0.90; Figs. S3 and S4). At the cellular level, single-cell transcriptomic analysis of human intestinal epithelial cells revealed an age-dependent decline in *ABCB1* expression (Fig. 1E), corroborated by pseudotime trajectory analysis showing progressive downregulation along differentiation states (Fig. 1F, G)[20].

Together, these findings indicate that intestinal P-gp efflux function declines with age, resulting in elevated systemic exposure to substrate drugs and increased risk of adverse outcomes (Fig. 1H). This age-associated reduction in intestinal P-gp highlights the need to investigate upstream regulatory factors, including the role of gut microbiota, in modulating P-gp expression and function.

### Effect of gut microbiota on intestinal P-gp expression

To determine whether aging reduces intestinal P-gp expression, we first analyzed the jejunum and ileum, key sites of drug absorption, in young (3 months, mus-y) and old (22–26 months, mus-o) C57BL/6 mice (n = 20). Both protein and mRNA levels of P-gp were significantly lower in the mus-o group than in the mus-y group, exhibiting consistent age-related reductions (Fig. 2A–F). Immunohistochemistry (IHC) further confirmed decreased membrane-localized P-gp in jejunal and ileal sections of aged mice (Fig. 2G–J).

Fecal samples were collected from young (3 months, mus-y), adult (12–15 months, mus-m), and old (22–26 months, mus-o) mice, as well as from young (18–30 years, HY) and older (over 75 years old, HO) humans. Differentiated human colon adenocarcinoma (Caco2) cell monolayers were incubated for 48 h with fecal water (FW) from each group. P-gp expression progressively declined in cells exposed to FW from mus-y, mus-m, and mus-o mice (Fig. 2K, L), and FW from HO humans significantly reduced both P-gp protein and *ABCB1* mRNA compared with HY controls (Fig. 2M–O).

To establish causality, 3-month-old C57BL/6 WT mice were colonized for 4weeks with fecal microbiota from either young (mus-yy) or old (mus-oy) donors (Fig. S5a). Compared with mus-yy recipients, mus-oy mice exhibited significantly lower P-gp protein and *Abcb1* mRNA expression in both jejunum and ileum (Fig. S5b-i). To confirm microbiota-dependent, mice pretreated with antibiotics (Abx) for 2 weeks were subsequently transplanted with microbiota from either mus-y (Abx-mus-yy) or mus-o (Abx-mus-oy) donors (Fig. 2P). Abx-mus-oy mice displayed markedly reduced intestinal P-gp expression (Figs. 2Q–V and S5j, k).

Collectively, these results demonstrate that aging-associated alterations in gut microbiota directly suppress intestinal P-gp expression, establishing a causal link between microbiota composition and transporter regulation.

### P-gp upregulation by *O. splanchnicus* and GDP-L-fucose

To identify specific gut microbes responsible for regulating intestinal P-gp, we profiled human fecal metagenomes (10 young adults, HY; 9 elderly, HO, Table S1) and murine 16S rRNA sequencing (10 juvenile, mus-y; 10 aged, mus-o). Both analyses revealed marked restructuring of microbial communities with age, as reflected by beta-diversity analyses at family, order, and genus levels (Figs. 3A–C, E, F and S6a–d). This decline was accompanied by consistent depletion of *Odoribacter*, particularly *O. splanchnicus*, in both species (Figs. 3D–L and S6e–f). Independent validation using PCR and qPCR confirmed a significant reduction of *O. splanchnicus* in the feces and small intestinal contents of aged mice (Fig. S6g–l). Cross-species prioritization thus highlighted *O. splanchnicus* as the most robust age-depleted taxon potentially linked to intestinal P-gp expression.

To evaluate its functional role, 3-month-old C57BL/6 mice were pretreated with antibiotics and subsequently colonized with either PBS (Abx-Control) or *O. splanchnicus* (Abx-OS) (Figs. 4A and S7a–d). Abx-OS mice exhibited significantly increased P-gp protein and *Abcb1* mRNA levels in jejunum and ileum, confirmed by Western blot, qPCR, and IHC (Figs. 4B–G and S7e–h). Similar restoration was observed in aged, antibiotic-pretreated mice gavaged with *O. splanchnicus* (Fig. S8a–f).

Untargeted metabolomic profiling of murine small intestine demonstrated that the relative abundance of GDP-L-fucose was markedly higher in young mice than in aged mice (log$_2$FC = 5.38), whereas other microbial metabolites, including short-chain fatty acids and related derivatives, declined modestly (Fig. 4H). KEGG pathway enrichment consistently highlighted fructose/mannose metabolism as the dominant age-associated pathway across human metagenomes, murine 16S data, and intestinal metabolomics (Figs. 4I, J and S9a, b). Key enzymes in this pathway, GDP-mannose 4, 6-dehydratase (GMDS, COG1089, and K01711) and GDP-L-fucose synthase (TSTA3 and K02377), were significantly enriched in young groups (Fig. 4K, L).

Survey of 1647 Human Microbiome Project genomes revealed that GMDS and TSTA3 are prevalent in *Bacteroidota*, including *O.*

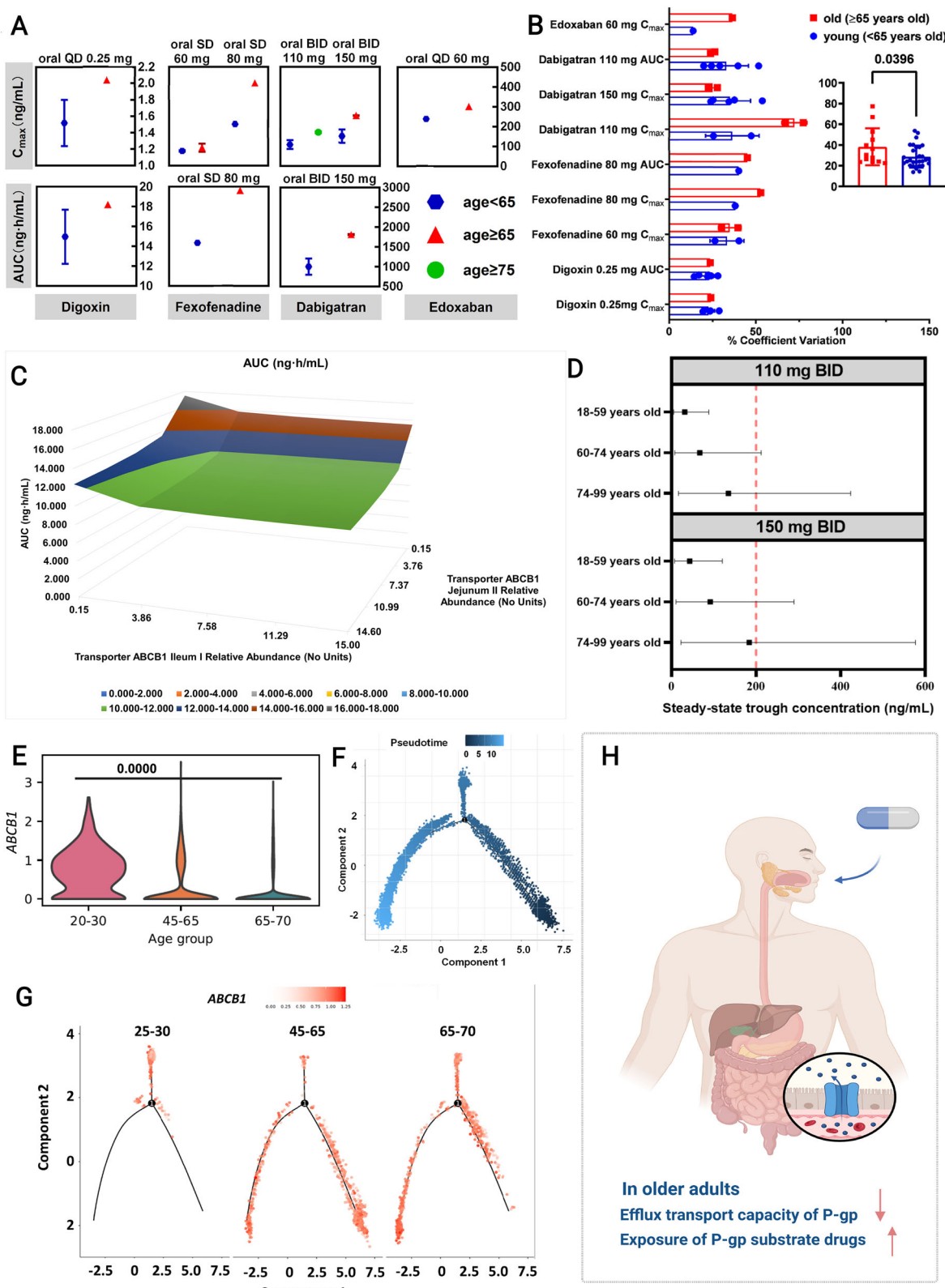

*splanchnicus* (Fig. S9c). Western blot and qPCR confirmed higher GMDS/TSTA3 expression in feces and small intestine contents of young vs aged mice (Fig. S9d–h). Colonization with *O. splanchnicus* increased GMDS/TSTA3 levels (Fig. S9i–n), and both enzymes were detectable in *O. splanchnicus* itself (Fig. S10a). Colonization with *O. splanchnicus* promoted GDP-L-fucose production, verified by LC–MS (Fig. S10b, c). *O. splanchnicus* abundance correlated positively with

Gmd/WcaG levels, supporting *an O. splanchnicus*–GMDS/TSTA3–P-gp regulatory axis (Fig. S10d–i).

Notably, GMDS and TSTA3 homologs were detected across multiple *Bacteroides* species within the *Bacteroidota* phylum, suggesting that this metabolic capacity may be shared by related commensals, although *O. splanchnicus* displayed one of the strongest age-associated declines and functional effects in our dataset.

**Fig. 1 | Aging-related decline in intestinal P-gp transport function results in an augmented exposure to substrate drugs and heightened safety concerns.**
**A** Comparison of drug exposure in older vs younger adults based on a systematic review of pharmacokinetic data for P-gp substrate drugs ($n = 14$/digoxin, $n = 8$/fexofenadine, $n = 18$/dabigatran, and $n = 2$/edoxaban). **B** Comparison of exposure variation in older vs younger adults based on systematic review of pharmacokinetic data for P-gp substrate drugs ($n = 9$/group). **C** Sensitivity analysis based on the influence of P-gp abundance in jejunum and ileum on dabigatran exposure.
**D** Comparison of pharmacokinetic profiles of dabigatran in the population across different age groups under identical dosing regimen (110 mg QD and 150 mg QD)

based on PBPK model ($n = 1000$). **E** Analysis of *ABCB1* single gene expression distribution in intestinal epithelial cells across different age groups ($n = 1024$/20–30 group, n = 2896/45–65 group, and n = 5732/65–70 group). **F** Pseudotime analysis of human intestinal epithelial cells ($n = 9652$). **G** Pseudotime analysis of human intestinal epithelial cells by different age groups ($n = 1024$/20–30 group, $n = 2896$/ 45–65 group, and $n = 5732$/ 65–70 group). **H** An overview of the scientific hypotheses of this study. Created in BioRender. Pmx, F. (2025) https://BioRender.com/ 1tz487n. Data are presented as mean ± SD. Statistical analyses were conducted using a two-tailed unpaired *t*-test in (**B**) and Mann–Whitney *U*-test in (**E**). Exact P values are reported in the figures.

Functional validation demonstrated that exogenous GDP-L-fucose (250 μM, 48 h) significantly upregulated P-gp expression in Caco2, LS180, and T84 cells (Figs. 4M, N and S11a–d). In vivo, antibiotic-pretreated mice gavaged with GDP-L-fucose (150 mg/L, 2 weeks) exhibited increased P-gp and *Abcb1* expression in jejunum and ileum (Figs. 4O–U and S11e–h).

Overall, these findings indicate that *O. splanchnicus* regulates intestinal P-gp expression via the GMDS/TSTA3-GDP-L-fucose axis. Age-related decline in *O. splanchnicus* and the associated biosynthetic pathway likely contributes to diminished P-gp function in older adults, providing mechanistic insight into microbiota-mediated regulation of intestinal drug transport.

### Engineered *E. coli* promotes intestinal P-gp expression in mice

Given that *O. splanchnicus* naturally produces GDP-L-fucose, we next asked whether forced overexpression of its biosynthetic pathway would be sufficient to induce intestinal P-gp expression. GDP-D-mannose is metabolized to GDP-L-fucose by GMDS (*Gmd*) and TSTA3 (*WcaG*). To functionally validate this pathway, we constructed a genetically engineered *E. coli* BL21 (DE3) strain carrying the pET-Gmd-WcaG plasmid (EGW), enabling GDP-L-fucose production (Figs. 5A and S12a–f). C57BL/ 6 WT mice (3 months old) were pretreated with antibiotics for 2 weeks and then colonized for an additional 2 weeks with PBS (Abx-Control), control-*E. coli* BL21 (Abx-EC), or EGW (Abx-EGW) (Fig. 5B). Successful colonization and GMDS and TSTA3 expression were confirmed in fecal DNA and protein extracts from Abx-EGW mice (Fig. S12g–l).

Functionally, intestinal P-gp protein and *Abcb1* mRNA levels were significantly elevated in both jejunum and ileum of Abx-EGW mice compared to controls (Fig. 5C–H), and IHC staining further confirmed increased membrane-localized P-gp (Fig. 5I, J).

Collectively, these results provide direct evidence that overexpression of GMDS and TSTA3 in engineered *E. coli* enhances GDP-L-fucose production and is sufficient to induce intestinal P-gp expression in vivo.

### GDP-L-fucose-mediated eIF4E phosphorylation activates c-Jun to promote P-gp expression

To elucidate the mechanism by which GDP-L-fucose enhances intestinal P-gp expression, we first performed RNA-seq analysis on Caco2 cells treated with GDP-L-fucose, *O. splanchnicus* supernatant, or PBS control. *ABCB1* mRNA was significantly upregulated in both GDP-L-fucose and *O. splanchnicus*-treated groups relative to controls (Fig. 6A). Correlation analysis of differentially expressed transcription factors indicated *JUN* and *MYC* as the strongest candidates associated with *ABCB1* expression (Fig. 6B). Because *MYC* was minimally expressed in mouse intestinal tissue, *JUN* was prioritized as the candidate regulator. In both intestinal samples and Caco2 cells, *JUN* expression correlated positively with *ABCB1* (Fig. S13a–l). ChIP-PCR and ChIP-qPCR further confirmed c-Jun binding to the *ABCB1* promoter (Fig. 6C–E).

To establish the functional role of *JUN*, Caco2 cells were transfected with *JUN*-targeting siRNA. Knockdown of *JUN* reduced both P-gp protein and *ABCB1* mRNA expression (Figs. 6F–H and S13m–o) and attenuated the stimulatory effects of GDP-L-fucose (Figs. 6I–K and S13p). These

results indicate that c-Jun is required for GDP-L-fucose-induced transcription of *ABCB1*.

Next, we investigated upstream mediators of this process. Structure-based docking simulations identified eIF4E and HPRT1 as candidate binding proteins for GDP-L-fucose (Table S2 and Figs. 7A, B and S14a, b). Microscale thermophoresis (MST) confirmed interactions of GDP-L-fucose with both proteins (Figs. 7C and S14c). However, overexpression experiments revealed that eIF4E, but not HPRT1, significantly increased c-Jun and P-gp expression (Figs. 7D, E and S14d, e). Consistently, GDP-L-fucose exposure increased phosphorylated eIF4E without altering total eIF4E levels (Fig. 7F, G). Silencing *EIF4E* abolished the induction of c-Jun and P-gp by GDP-L-fucose (Fig. 7H–I).

Together, these findings demonstrate that GDP-L-fucose binds to eIF4E and promotes its phosphorylation, which in turn activates c-Jun and enhances *ABCB1* transcription, thereby increasing intestinal P-gp expression.

## Discussion

Aging is associated with a progressive decline in intestinal P-gp activity, thereby heightening susceptibility to drug-drug interactions and adverse events in older adults[21,22]. In this study, we identify *O. splanchnicus* as a microbiota-derived regulator of P-gp that operates through the GMDS/TSTA3-GDP-L-fucose axis to sustain transporter expression (Fig. 8). Both human and murine datasets consistently demonstrated age-related depletion of *O. splanchnicus*, reduced microbial capacity for GDP-L-fucose biosynthesis, and diminished intestinal P-gp levels. Functional experiments further confirmed that supplementing *O. splanchnicus* or GDP-L-fucose was sufficient to rescue P-gp expression in intestinal tissues and cell models, thus establishing a mechanistic link between microbial aging and host drug transport function. We also observed that colonic P-gp expression declines with aging and can be restored by young microbiota, *O. splanchnicus*, GDP-L-fucose, or engineered bacteria (Fig. S15). Importantly, the regulatory patterns observed in the colon are consistent with those in the small intestine, where P-gp also declines with age. From a pharmacological perspective, small intestinal P-gp plays a predominant role in systemic drug absorption and the risk of ADRs, underscoring its critical importance in maintaining drug homeostasis[23].

Traditional approaches to assessing intestinal P-gp include tissue biopsies for expression analysis and probe substrate pharmacokinetics for functional evaluation. While informative, both approaches are limited. Biopsies suffer from spatial sampling heterogeneity and small sample sizes, whereas pharmacokinetics are confounded by multisystem changes that accompany aging. Indeed, one human study reported an approximate 60% reduction in intestinal *ABCB1* expression with aging ($n = 8$), but interpretation was constrained by small sample size and spatial bias[24]. To complement these approaches, we analyzed publicly available single-cell RNA-seq datasets, which independently confirmed age-related decreases in *ABCB1* expression[20]. However, expression-based analyses alone cannot capture transporter function. To address this gap, we employed PBPK modeling using dabigatran etexilate as a probe substrate. This retrospective analysis predicted reduced intestinal P-gp activity in older adults[25]. Together, these complementary approaches, biopsy, transcriptomic, pharmacokinetic,

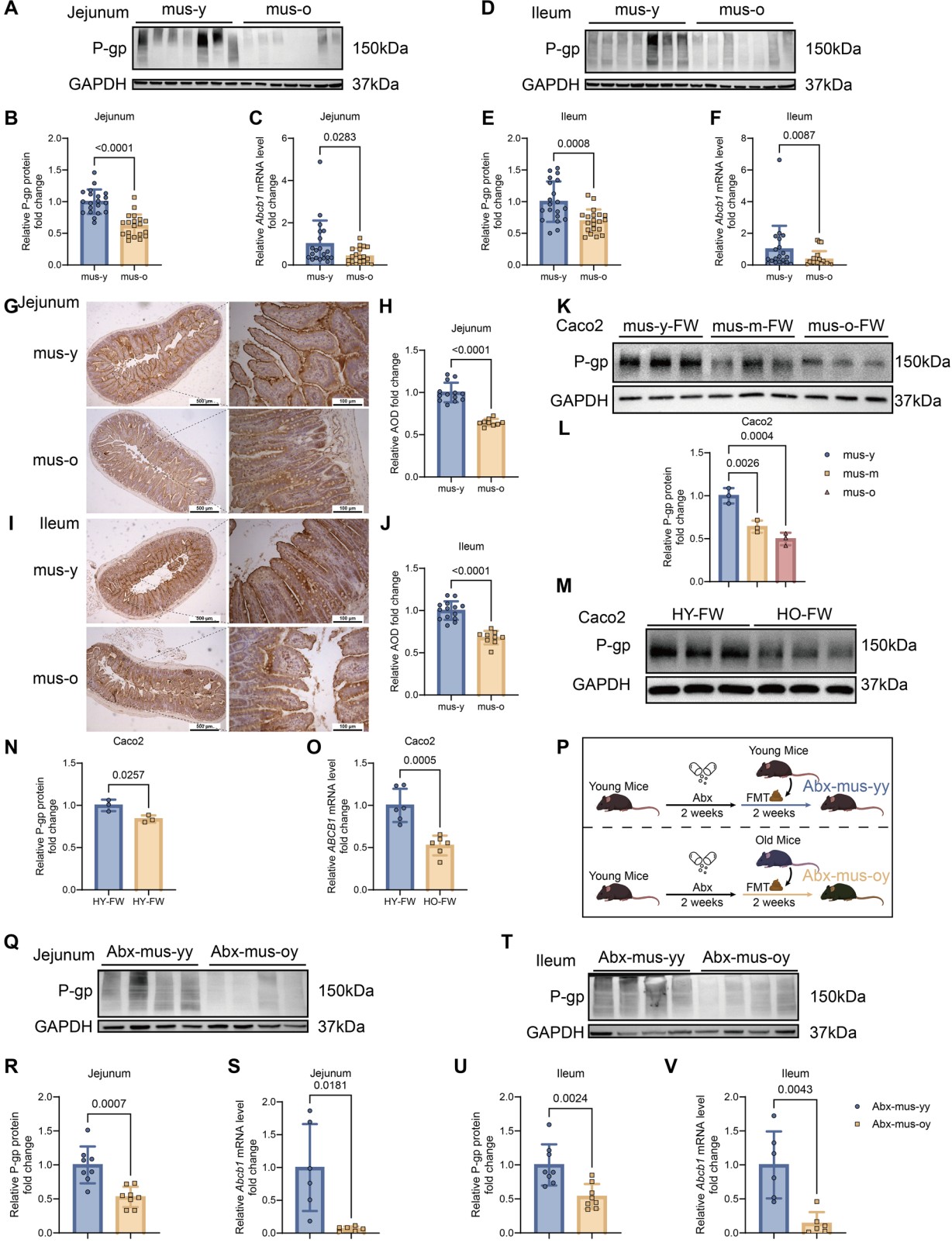

and modeling, consistently support the conclusion that intestinal P-gp declines with aging, providing both mechanistic and translational evidence for altered drug disposition in older adults.

Although gut microbiome dysbiosis has been broadly implicated in age-related diseases, its influence on xenobiotic transporters has received little attention[26–29]. Previous studies showed that colonization with *Bacteroides thetaiotaomicron* can regulate intestinal gene

expression and that members of the *Clostridia* and *Bacilli classes* are required for P-gp induction[3,30]. In line with these observations, we found that higher-order taxa such as *Bacteroideaceae*, *Bacteroidales*, and *Bacteroidetes* were reduced in older individuals, consistent with previous reports[31,32]. Importantly, our cross-species analyses identified *O. splanchnicus* as the most consistently depleted taxon with aging, whereas ten other representative *Bacteroides* and *Parabacteroides*

**Fig. 2 | The expression of intestinal P-gp significantly declines in aged mice, closely associated with gut microbiota changes, and has been validated using human intestinal cells. A, B** Protein expression levels of P-gp in the jejunum of young (3-month-old, mus-y) and old (22–26-month-old, mus-o) mice were measured by western blotting (**A**), and relative expression was quantified (**B**) ($n = 20$/group). **C** *Abcb1* mRNA levels in the jejunum of mus-y and mus-o groups ($n = 20$/group). **D, E** P-gp expression in the ileum was detected by western blot (**D**), and relative levels were analyzed (**E**) ($n = 20$/group). **F** *Abcb1* mRNA levels in the ileum of mus-y and mus-o groups ($n = 20$/group). **G, H** Representative IHC staining of P-gp in jejunum sections (scale bar: 500 and 100 μm) and quantitative analysis of P-gp positive areas ($n = 13$ and 9/group). **I, J** IHC staining of P-gp on ileum sections (scale bar: 500 and 100 μm) and analysis of positive areas ($n = 14$ and 9/group). **K, L** P-gp expression in Caco2 cells after 48 h exposure to fecal water (FW) from young (mus-y-FW), middle-aged (12–15-month-old, mus-m-FW), or old (mus-o-FW) mice (**K**),

with relative quantification (**L**) ($n = 3$/group). **M, N** P-gp expression in Caco2 cells treated for 48 h with FW from young human adults (22-year-old, HY-FW) or elderly adults (over 75-year-old, HO-FW) (**M**), and relative protein levels (**N**) ($n = 3$/group). **O** *ABCB1* mRNA expression in Caco2 cells after exposure to HY-FW or HO-FW ($n = 6$/group). **P–V** Twelve-week-old C57BL/6 WT mice were treated with Abx for 2 weeks, followed by fecal microbiota transplantation (FMT) from young (Abx-mus-yy) or old (Abx-mus-oy) mice for another 2 weeks. **P** Schematic overview of the animal experiment. **Q, R** P-gp expression in the jejunum (**Q**) and relative quantification (**R**) of Abx-mus-yy and Abx-mus-oy groups ($n = 8$/group). **S** *Abcb1* mRNA levels in the jejunum ($n = 6$/group). **T, U** P-gp expression in the ileum (**T**) and relative quantification (**U**) ($n = 8$/group). **V** *Abcb1* mRNA expression in the ileum ($n = 6$/group). Data are presented as mean ± SD. Statistical analyses were conducted using two-tailed unpaired *t*-test (**B, J, N, O, R, U**), Welch's corrected *t*-test (**E, H, S**), Mann–Whitney *U*-test (**C, F, V**), and one-way ANOVA (**L**). Exact *P* values are reported in the figures.

species showed no significant differences between young and old mice or humans (Fig. S16). This specificity suggests that the decline of *O. splanchnicus*, rather than the general loss of *Bacteroides*, underlies the reduction in GDP-L-fucose biosynthesis and P-gp activity.

*O. splanchnicus* is a beneficial gut commensal whose abundance declines with age and deteriorating health, a trend we observed in our study and that was independently validated in a large cohort of 10,207 individuals aged 40–93 years with follow-up data[33]. Notably, it remains enriched in healthy older adults and centenarians, suggesting a potential role in resilience against age-related physiological decline[34–38]. Its abundance is associated with favorable metabolic and intestinal profiles, and depletion has been linked to inflammatory and metabolic disorders, including inflammatory bowel disease (IBD), fatty liver disease, and colon cancer[39,40]. The protective functions of *O. splanchnicus* are multifaceted. Beyond metabolite-mediated regulation of host xenobiotic transporters, it can directly suppress enteric pathogens through bacteriocin secretion, such as against Salmonella, highlighting its contribution to gut homeostasis[41].

Importantly, while *O. splanchnicus* is largely beneficial, isolated reports have linked it to pathological conditions, including cardiovascular, renal, and intracranial diseases, likely reflecting opportunistic behavior under severe inflammation or compromised intestinal barriers[42]. These findings underscore the need for careful safety evaluation in therapeutic applications, but overall support its role as a key protective commensal whose age-related depletion may impair intestinal homeostasis and drug detoxification.

GDP-L-fucose plays essential roles in tissue development, angiogenesis, inflammation, and cancer progression[43]. In our study, we identify it as a key mediator linking *O. splanchnicus* to P-gp regulation in older adults. Although host cells can synthesize GDP-L-fucose via GMDS and TSTA3, our data show that expression of these enzymes remains stable across age groups and is unaffected by microbiota interventions (Figs. S17 and 18)[44]. This indicates that age-related decline in GDP-L-fucose availability primarily arises from microbial, rather than host, factors. From a translational perspective, our findings highlight two complementary strategies to enhance intestinal P-gp activity. One approach is to modulate the gut microbiota, for example, through *O. splanchnicus* probiotics or fecal microbiota transplantation (FMT). Another strategy is to directly supplement the key metabolite, GDP-L-fucose, or its analogs. These interventions can be tailored to individual patient factors, including comorbidities, polypharmacy, and microbiome composition, enabling precision geriatric therapeutics[42,45,46].

Although *O. splanchnicus* exhibited a reproducible age-related decline and a clear capacity to restore intestinal P-gp expression, we acknowledge that GMDS and TSTA3 are not unique to this species. Comparative genomic analysis revealed that homologs of these enzymes are broadly distributed among members of the *Bacteroidota* phylum. In our study, *O. splanchnicus* was selected as a representative commensal with a strong and consistent negative correlation with age,

and whose colonization robustly increased GMDS/TSTA3 expression and GDP-L-fucose production. While our data demonstrate that *O. splanchnicus* can rescue P-gp dysfunction through the GMDS/TSTA3-GDP-L-fucose axis, we do not exclude the possibility that other *Bacteroides*-related taxa may exert similar effects. Further comparative colonization or co-culture experiments using additional *Bacteroides* strains will be required to systematically determine the extent to which this mechanism is conserved across the phylum.

Together, these findings suggest that age-related depletion of multiple *Bacteroides* species may contribute to diminished intestinal fucosylation and P-gp function, with *O. splanchnicus* representing one key mediator within this broader microbial network.

## Methods
### Human sample collection
Stool samples from adults aged ≥75 years were collected in a non-interventional clinical trial (nos. M2021373; ChiCTR2100054184). Samples from adults aged 18–30 years were obtained from a previously published trial (Peking University Third Hospital Ethics Committee approval: M2020390; NCT04743726)[47]. All participants provided informed consent. Samples were flash-frozen and stored at −80 °C. All the stool samples from these two studies were processed and analyzed in the same batch under identical laboratory conditions to ensure consistency.

### Systematic review of pharmacokinetic data for P-gp substrates
We searched PubMed (up to August 2024) for clinical studies reporting pharmacokinetics of FDA-defined P-glycoprotein (P-gp) substrates (dabigatran etexilate, digoxin, edoxaban, fexofenadine) in young and older adults. Keywords were predefined, and studies were eligible if they stratified pharmacokinetic outcomes (AUC, $C_{max}$) by age and reported sample size, demographics, dosing, and variability. In total, we identified seven studies on digoxin, three on fexofenadine, six on dabigatran, and two on edoxaban. Owing to the limited number and heterogeneity of such studies, no meta-analysis was performed. Instead, parameters were descriptively summarized and compared across age groups using GraphPad Prism v8.0. The "*n*" shown in Fig. 1A, B directly reflects the sample sizes of the original studies. Across all available reports, older adults consistently exhibited higher systemic exposure to P-gp substrates, supporting the hypothesis that age-related reductions in P-gp function contribute to altered drug disposition.

### Meta-analyses of bleeding risk with DOACs
To evaluate clinical correlates of P-gp function, we conducted two independent meta-analyses of RCTs involving DOACs (dabigatran etexilate, edoxaban, rivaroxaban, and apixaban).

**Age-related bleeding risk.** A comprehensive search of the FDA drug database and ClinicalTrials.gov was performed to identify all published

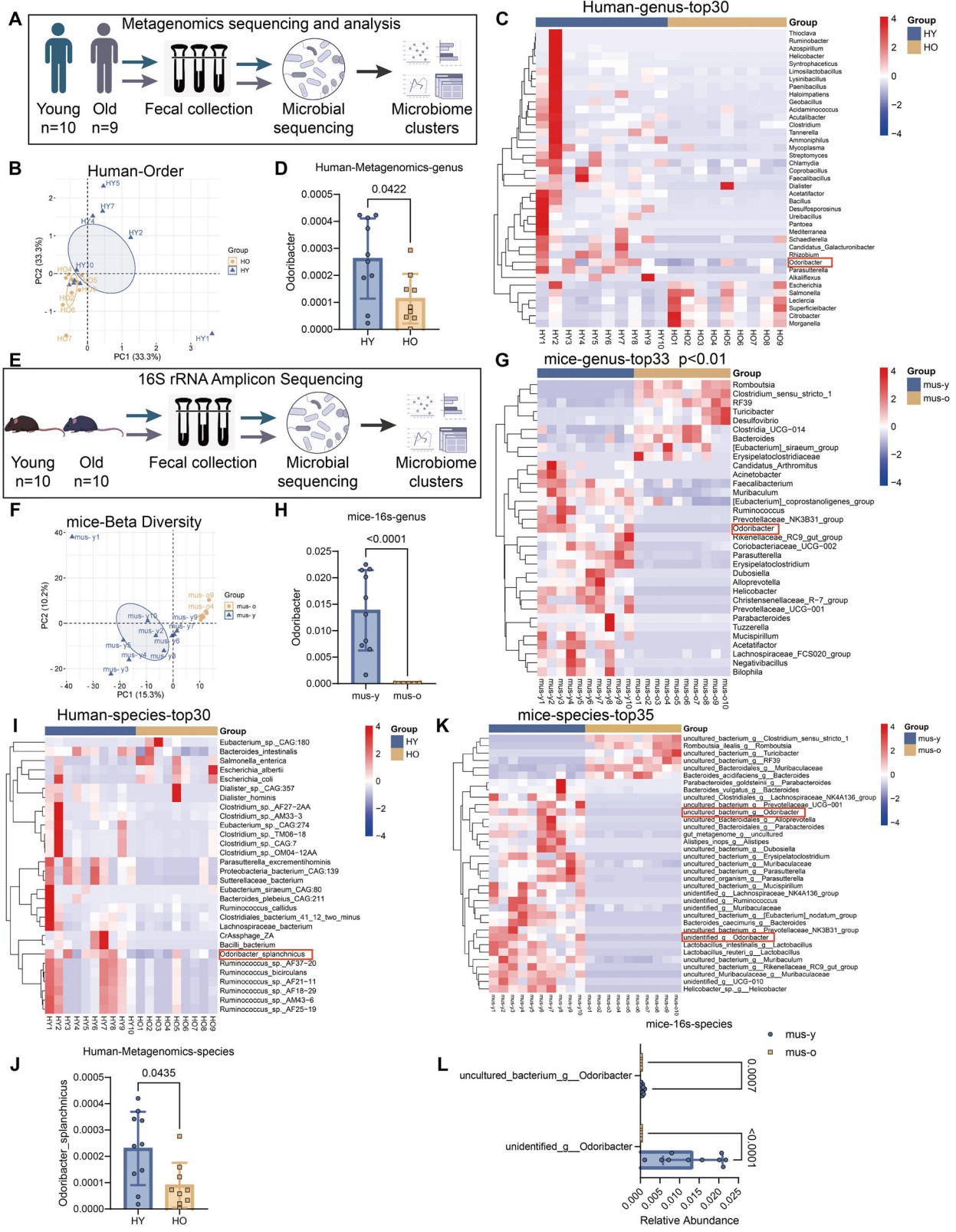

phase III RCTs of FDA-approved DOACs. Exclusion criteria were: (i) ANDA drugs, (ii) non–phase III trials, (iii) patient age < 18 years, (iv) non-oral administration, (v) lack of randomization, (vi) absence of age-stratified outcomes, (vii) multi-drug regimens, and (viii) non-targeted drugs. After applying these criteria, 12 eligible trials were included. The primary endpoint was major bleeding, compared between younger and older age groups across different DOAC doses.

**Impact of concomitant P-gp inhibitors**. The same search and screening strategy was applied, with the additional requirement of reporting concomitant P-gp inhibitor use. After exclusions, two RCTs (dabigatran etexilate and edoxaban) were included. Major bleeding events were compared between patients treated with or without P-gp inhibitors under single-agent DOAC therapy.

**Fig. 3 | *O. splanchnicus* is under-represented in the gut microbiome of both aged humans and mice. A**–**D** The metagenomics gene profiling data for fecal microbiome from HY and HO groups (*n* = 10 and 9/group). **A** Overview of metagenomic sequencing experiments on HY and HO feces. **B** PCoA plot of beta-diversity at the order level. **C** Relative abundance of significantly altered taxa at the rank of genus (including unspecified taxa). **D** Comparison of the relative abundance of *Odoribacter* at the genus level (*n* = 10 and 9/group). **E**–**H** The 16S rRNA gene profiling data for fecal microbiome from mus-y and mus-o groups (*n* = 10/group). **E** Overview of 16S rRNA sequencing experiments on mus-y and mus-o feces. **F** PCoA plot of bacterial beta-diversity. **G** Relative abundance of significantly altered taxa at the rank of genus (including unspecified taxa). **H** Comparison of the relative abundance of *Odoribacter* at the genus level (*n* = 10/group). **I**, **J** The metagenomics gene profiling data for fecal microbiome from HY and HO groups (*n* = 10 and 9/group). **I** Relative abundance of significantly altered taxa at the rank of species (including unspecified taxa). **J** Comparison of the relative abundance of *O.splanchnicus* at the species level. **K**, **L** The 16S rRNA gene profiling data for fecal microbiome from mus-y and mus-o groups (*n* = 10/group). **K** Relative abundance of significantly altered taxa at the rank of species (including unspecified taxa). **L** Comparison of the relative abundance of unidentified_g_*Odoribacter* and uncultured_bacterium_g_*Odoribacter* at species level. Data are presented as mean ± SD. Statistical analyses were conducted using Mann–Whitney *U*-test (**D**, **H**, **J**, **L**). Exact *P* values are provided in the figures.

For both analyses, individual study data were pooled using the Mantel–Haenszel (M–H) fixed-effect model. Risk ratio (RR) with its corresponding 95% confidence interval (CI) was calculated as a combined effect measure. Statistical heterogeneity of the overall or subgroup was calculated using the Chi-square test and quantified using I2. An I2 value larger than 50% revealed that there was substantial heterogeneity. The statistical significance of the overall effect was assessed by a Z-test, with a *P* value less than 0.05 considered statistically significant. All analyses were conducted using Review Manager (Rev-Man, v5.3, Cochrane Collaboration, UK).

## PBPK modeling and simulation
A PBPK model of dabigatran etexilate and its active metabolite dabigatran, was established in Simcyp® Simulator (v22, Certara, UK) based on our previous validated work[18]. Virtual populations were parameterized using our Chinese older-subject PBPK database[25]. Sensitivity analyses quantified the contribution of intestinal P-gp to dabigatran systemic exposure. For each subgroup, 1000 virtual individuals (10 trials, 100 subjects each) were simulated. Model predictions were compared with observed clinical data to estimate age-related reductions in intestinal P-gp abundance and their impact on dabigatran exposure.

## Single-cell transcriptomic analysis
Single-cell RNA sequencing data of human intestinal epithelial cells across developmental and aging stages were obtained from the Gut Cell Atlas (https://www.gutcellatlas.org/spacetime/epithelium/) published by Elmentaite et al.[20]. Data were processed in R (v4.3.1). Cell type annotation, quality control, and differential expression were performed using the omicverse package (v1.6.10), with significance defined as adjusted *p* < 0.05. Pseudotime analysis was conducted with Monocle2 to reconstruct epithelial differentiation trajectories and assess *ABCB1* expression dynamics.

## Animal experiments
Animal experiments were approved by the Institutional Animal Care and Use Committee of Peking University and conducted in accordance with NIH guidelines. C57BL/6 mice were sourced from the Laboratory Animal Center of Peking University Health Science Center and Beijing Vital River Laboratory Animal Technology Co., Ltd.

The mice were housed under controlled conditions at a temperature of 21 ± 1 °C and a relative humidity of 50% ± 5%, following a 12-h light/dark cycle. They had ad libitum access to standard mouse feed and water throughout the experiments. Animal welfare was strictly ensured, and all procedures adhered to ethical regulations established by the European Parliament for the protection of animals used in scientific research.

## Mice intestinal tissue acquisition
Before necropsy, mice were fasted for a minimum of 4 h and anesthetized with ketamine-xylazine or isoflurane. Blood was collected from the ocular vein using a blood collection vessel. Phosphate-buffered saline (PBS) was used to remove intestinal tissue and cleanse surrounding adipose tissue. The jejunum (1–2 cm distal to the stomach) and ileum (1–2 cm proximal to the cecum) were excised, rinsed with cold PBS. Intestinal tissues were rapidly retrieved and immediately frozen in liquid nitrogen before storage at −80 °C for subsequent analysis.

## FMT
Fecal samples were collected from healthy C57BL/6 mice aged 3 months and 22–26 months. After thorough mixing at room temperature, the samples were diluted in PBS to create an FMT suspension with a concentration of 400 mg/mL. Subsequently, the fecal suspension was centrifuged, and the supernatant was extracted to obtain the fecal graft.

The experimental group of 3-month-old mice was randomly divided into two subgroups. These mice received fecal grafts from either 3-month-old or 22- to 26-month-old mice via gavage (oral administration) at a dosage of 200 μL per mouse, once daily for 4 weeks. Intestinal tissues from the experimental group were collected for subsequent analysis.

## Antibiotic supplementation
An antibiotic cocktail (Abx) was administered to the mice. The Abx consisted of the following components:

Vancomycin (V871983-1g, Macklin) at a concentration of 0.5 g/L

Metronidazole (M813526-100g, Macklin) at a concentration of 0.5 g/L

Neomycin sulfate (N6063-25g, Macklin) at a concentration of 1 g/L

Ampicillin trihydrate (A800200-25g, Macklin) at a concentration of 1 g/L

The Abx was provided via drinking water and renewed every other day. In the adequate treatment groups, Abx was administered continuously for two weeks to ensure sufficient microbial depletion.

The 3-month-old mice were randomly divided into two groups. Both groups received a 2-week treatment with the Abx. Fecal grafts from both 3-month-old and 22- to 26-month-old mice were administered via gavage (200 μL each) once a day for 2 weeks. Subsequently, intestinal tissues from the experimental group were collected for follow-up analysis.

## Metagenomics sequencing and analysis
Total microbial DNA was extracted from human fecal samples using the DNeasy Power Soil Kit (QIAGEN, Germany). Nineteen DNA samples (10 from young normal individuals and 9 from old normal individuals) passed quality control. Whole-genome shotgun metagenome sequencing was performed on the HiSeq 4000 platform (Illumina) with paired-end reads of 150 base pairs at Shanghai OE Biotech Co., Ltd. Raw sequencing data underwent processing using the Trimmomatic V0.36 tool, including adapter trimming and removal of low-quality reads or base pairs. Bowtie 2 was used to eliminate host contaminations by aligning against the reference human genome (version hg38). DIAMOND software facilitated comparison with the NCBI NR library to identify proteins with high sequence similarity for functional annotation. Clean reads were

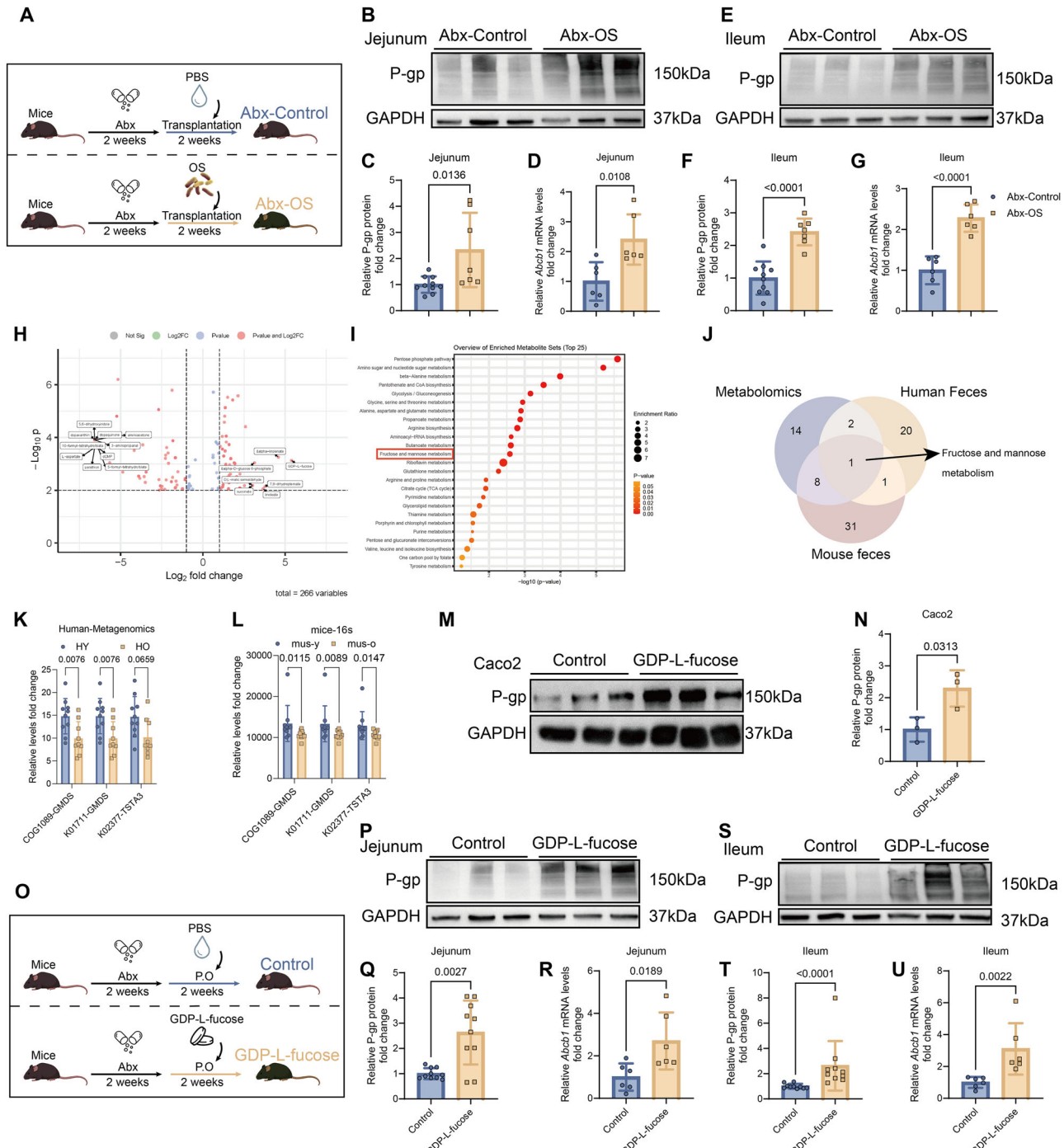

**Fig. 4 | *O. splanchnicus* and GDP-ʟ-fucose promote P-gp expression in both the jejunum and the ileum of mouse models. A–G** Twelve-week-old C57BL/6 WT mice were treated with Abx for 2 weeks, followed by colonization with PBS (Abx-Control) or OS (Abx-OS) for 2 weeks. **A** Schematic of the Abx-Control and Abx-OS experimental design. **B, C** P-gp expression in jejunum (**B**) and relative quantification (**C**) of Abx-Control and Abx-OS groups (*n* = 10 and 7/group). **D** *Abcb1* mRNA levels in jejunum (*n* = 6/group). **E, F** P-gp expression in ileum (**E**) and relative quantification (**F**) of Abx-Control and Abx-OS groups (*n* = 10 and 7/group). **G** *Abcb1* mRNA levels in ileum (*n* = 6/group). **H, I** Untargeted metabolomics analysis performed on small intestine from mus-y and mus-o groups (*n* = 6/group). **H** Volcano plots of differential metabolites. **I** KEGG pathway enrichment analysis of differential metabolites (MetaboAnalyst 5.0). **J** Venn diagram of **I** and Figs. S9a, b (mouse intestinal tissue, human fecal, and mouse fecal). **K, L** Comparison of relative expression levels of GMDS (COG1089, K01711) and TSTA3 (K02377). **K** Relative expression levels of GMDS (COG1089, K01711) and TSTA3

(K02377) in the metagenomics gene profiling data for fecal microbiome from HY and HO groups (*n* = 10 and 9/group). **L** Relative expression levels of GMDS (COG1089, K01711) and TSTA3 (K02377) in the 16S rRNA gene profiling data for fecal microbiome from mus-y and mus-o groups (*n* = 10/group). **M, N** P-gp expression (**M**) and relative quantification (**N**) in Caco2 cells treated with PBS or GDP-ʟ-fucose (250 µM, 48 h) (*n* = 3/group). **O–U** Twelve-week-old C57BL/6 WT mice received Abx for 2 weeks, then were administered with or without GDP-ʟ-fucose (150 mg/L) for 2 weeks. **O** Experimental outline of Control and GDP-ʟ-fucose groups. **P, Q** P-gp expression in jejunum (**P**) and relative quantification (**Q**) (*n* = 10/group). **R** *Abcb1* mRNA levels in jejunum (*n* = 6/group). **S, T** P-gp expression in ileum (**S**) and relative quantification (**T**) (*n* = 10/group). **U** *Abcb1* mRNA levels in ileum (*n* = 6/group). Data are presented as mean ± SD. Statistical analyses were conducted using two-tailed unpaired *t*-test (**F, G, N, R**), two-tailed unpaired *t*-test with Welch's correction (**Q**), and Mann–Whitney *U*-test (**C, D, K, L, T, U**). Exact *P* values are reported in the figures.

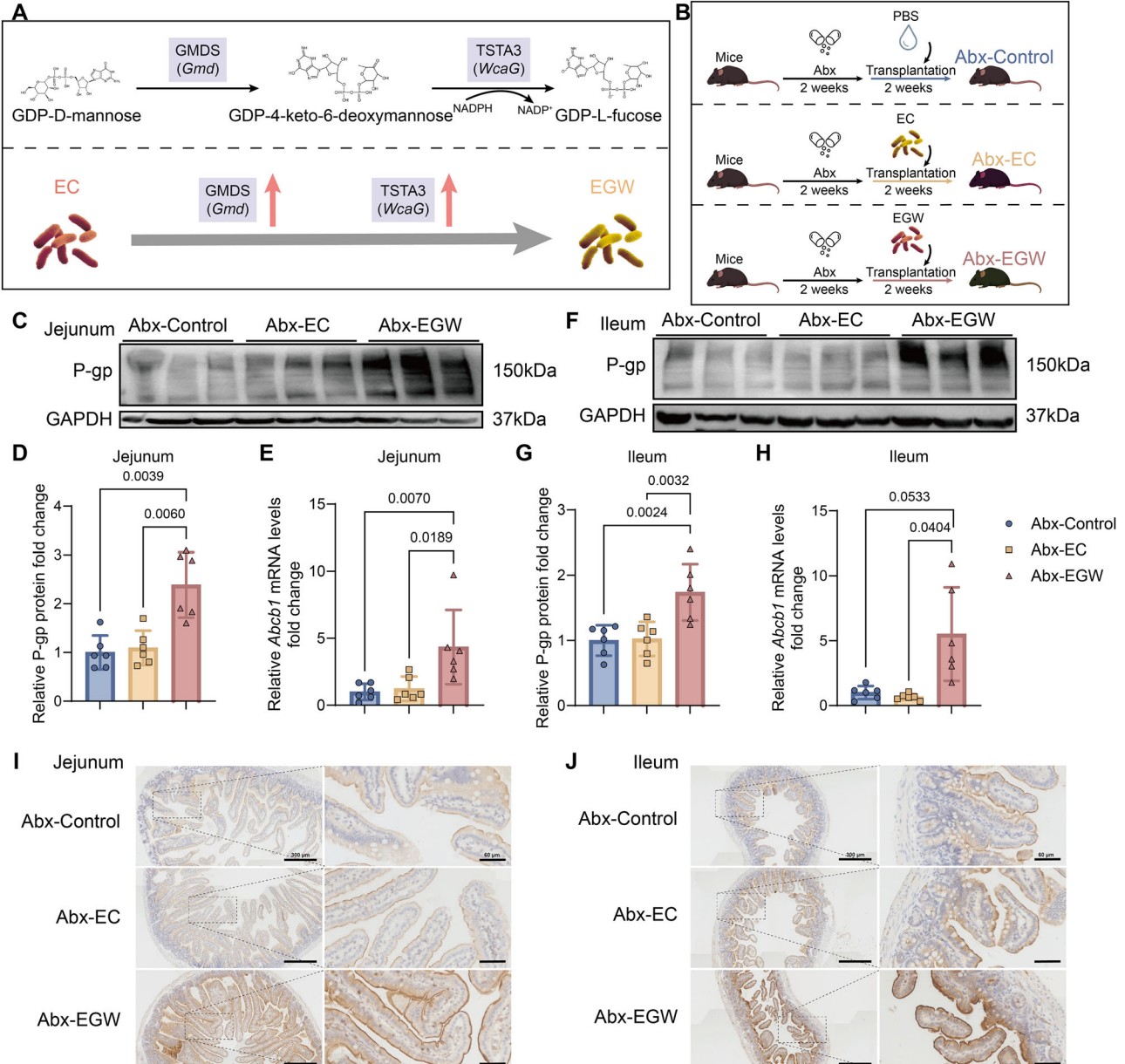

**Fig. 5 | The overexpression of GMDS and TSTA3 in *E. coli* leads to an increase in intestinal P-gp expression. A** Overview of constructing engineered bacterial EGW overexpressing *Gmd* and *WcaG* from BL21 (DE3) (E-Control, EC). **B** Overview of animal experiments of Abx-Control, Abx-EC, and Abx-EGW mouse models. **C, D** Expression levels of P-gp in jejunum (**C**) and relative quantification (**D**) of Abx-Control, Abx-EC, and Abx-EGW groups (*n* = 6/group). **E** *Abcb1* mRNA expression levels in the jejunum of Abx-Control, Abx-EC, and Abx-EGW groups (*n* = 6/group).

**F, G** Expression levels of P-gp in ileum (**F**) and relative quantification (**G**) of Abx-Control, Abx-EC, and Abx-EGW groups (*n* = 6/group). **H** *Abcb1* mRNA expression levels in the ileum of Abx-Control, Abx-EC, and Abx-EGW groups (*n* = 6/group). **I, J** IHC staining of P-gp on jejunum (**I**) and ileum (**J**) sections (Abx-Control, Abx-EC, and Abx-EGW groups, scale bar: 300 and 60 μm). Data are presented as mean ± SD. Statistical analyses were conducted using one-way ANOVA (**D, E, G, H**). Exact *P* values are reported in the figures.

---

further analyzed for taxonomic profiling using default parameters on the OE Cloud Platform.

### 16S rRNA amplicon sequencing

Microbial DNA was extracted from mouse fecal samples using the DNeasy Power Soil Kit (QIAGEN, Germany). The V3 and V4 regions of the 16S rRNA gene sequence in mice fecal microbiota were determined by high-throughput sequencing analysis with the forward primer of 343F (5′-TACGGRAGGCAGCAG-3′), and the reverse primer of 798R (5′-AGGGTATCTAATCCT-3′). High-throughput sequencing analysis was performed using the Illumina MiSeq platform. The resulting data were analyzed on the OE Cloud Platform (Shanghai OE Biotech Co., Ltd, Shanghai, China).

### Untargeted mass spectrometry for metabolomics analysis

For sample preparation, we accurately weighed 30 mg of tissue sample into a 1.5 mL Axygen tube. Next, we added 400 μL of a methanol: acetonitrile: aqueous solution (*v:v:v* = 2:2:1). The sample was pre-cooled at −20 °C for 2 min and then ground using a grinder (60 Hz, 2 min). Afterward, we subjected it to an ice water bath ultrasound for 10 min followed by another cooling step at −20 °C for 30 min. Finally, we centrifuged the sample for 10 min (13,000 rpm, 4 °C) and collected 250 μL of the supernatant.

The untargeted metabolomics analysis was performed using a Q Exactive HFX Orbitrap mass spectrometer (Thermo, CA). We loaded one microliter of the supernatant onto a normal-phase chromatography column, eluting it to the mass spectrometer with 50%

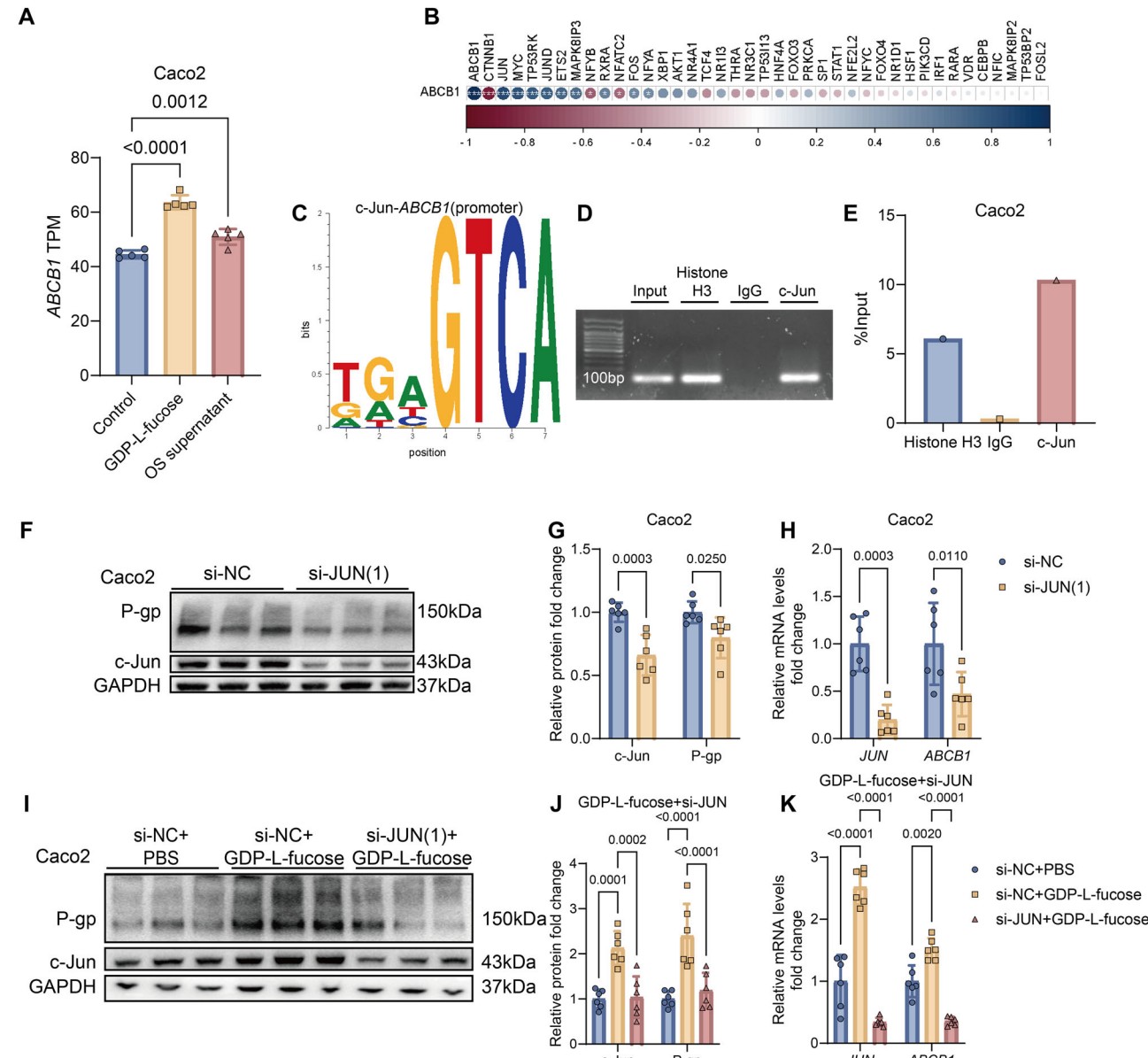

**Fig. 6 | GDP-ʟ-fucose induces *ABCB1* transcription via c-Jun to enhance the expression of Caco2 cells. A**, **B** Whole RNA-seq analysis of control Caco2 cells (PBS) or cells exposed for 48 h to GDP-ʟ-fucose (250 μM), or to *O.splanchnicus* (OS) supernatant (*n* = 5/group). **A** TPM levels of *ABCB1* in the three groups. **B** Spearman's correlation analysis between the expression of *ABCB1* and each of the transcription factors (TFs) known or predicted to have inductive, variable, or repressive effects on *ABCB1* expression. **C** Predicted c-Jun binding *ABCB1* promoter motifs by UCSC Genome Browser Home and PROMO. **D**, **E** The binding of *ABCB1* promoter and Histone H3, IgG, and c-Jun was further explored by ChIP-PCR (**D**) and ChIP-qPCR (**E**) assays in Caco2 (*n* = 6/group). **F**–**H** Caco2 cells were transfected with si-NC or si-JUN (1) for 48 h (*n* = 6/group). **F**, **G** Expression levels of P-gp and c-Jun (**F**) and relative quantification (**G**) in Caco2 cells. **H** *ABCB1* and *JUN* mRNA levels in Caco2 cells. **I**–**K** Caco2 cells were transfected with si-NC or si-JUN (1) and with or without GDP-ʟ-fucose (250 μM) for 48 h (*n* = 6/group). **I**, **J** Expression levels of P-gp and c-Jun (**I**) and relative quantification (**J**) in Caco2 cells. **K** *ABCB1* and *JUN* mRNA levels in Caco2 cells. Data are presented as mean ± SD. Statistical analyses were conducted using one-way ANOVA (**A**) and two-way ANOVA (**G**, **H**, **J**, **K**). Exact *P* values are reported in the figures.

acetonitrile containing 10 mM ammonium acetate as the eluent. Data were acquired in both positive and negative ion modes with data-dependent MS/MS acquisition. The full scan and fragment spectra were collected with resolutions of 70,000 and 17,500, respectively.

The source parameters were as follows:
Spray voltage: 3000 V
Capillary temperature: 320 °C
Heater temperature: 300 °C
Sheath gas flow rate: 35
Auxiliary gas flow rate: 10

Metabolite identification relied on a TraceFinder search using a home-built database. We performed PCA and orthogonal partial least squares discriminant analysis (PLS-DA) on all data using the online platform MetaboAnalyst 5.0. To assess changes in metabolite levels between the mus-y and mus-o groups, we conducted the Wilcoxon test in R-3.6.3. Significant results were defined as FDR < 0.05 and absolute | log$_2$ (fold change)|>1. Additionally, MetaboAnalyst 5.0 facilitated pathway enrichment analysis based on potential metabolites.

### *O. splanchnicus* culture and transplantation

Dry powder of *O. splanchnicus* (ATCC29572, BNCC359789, North Na Biological) was cultured on Columbia blood agar plates (CA-B) at 37 °C for 3–5 days under anaerobic conditions to obtain a single colony. *O. splanchnicus* was routinely cultured in brain heart infusion broth (BHI)

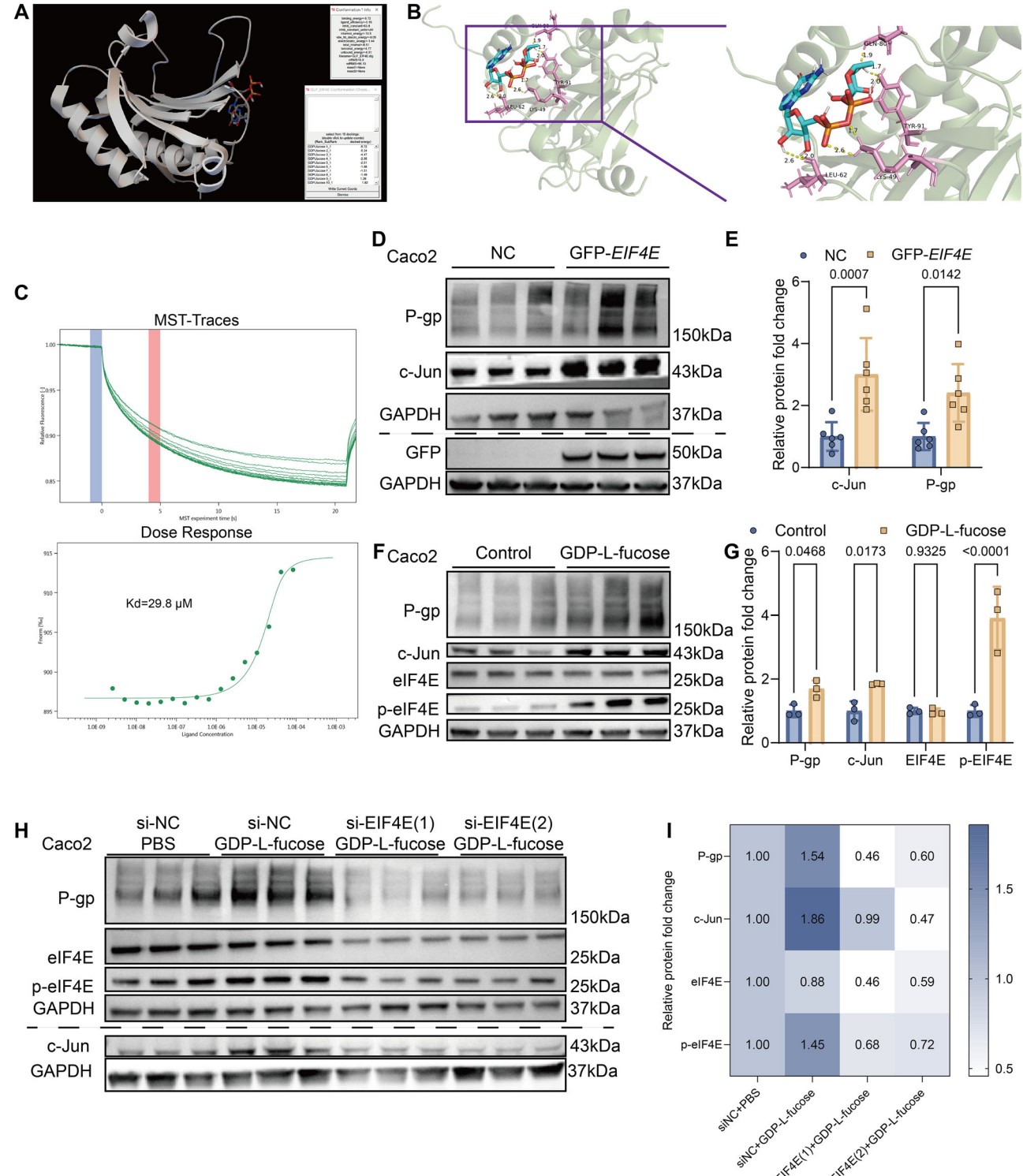

**Fig. 7 | GDP-L-fucose activates c-Jun via promoting phosphorylation of eIF4E to enhance the P-gp expression of Caco2 cells. A** AutoDock4.2 was utilized to achieve molecular docking between GDP-L-fucose and eIF4E. **B** Left—general view of docked GDP-L-fucose into eIF4E. Right—zoomed-in region. The protein is a green and pink cartoon outfit. The pink part represents the name of the specific amino acid to which GDP-L-fucose binds to eIF4E, and the length and number of hydrogen bonds. **C** Binding of GDP-L-fucose to eIF4E as analyzed by MST. **D**, **E** Expression levels of GFP, c-Jun, and P-gp (**D**) and relative quantification (**E**) in Caco2 cells

transfected with NC or GFP-eIF4E plasmid 48 h ($n$ = 6/group). **F**, **G** Expression levels of p-eIF4E, eIF4E, c-Jun, and P-gp (**F**) and relative quantification (**G**) in Caco2 cells exposed to control medium or GDP-L-fucose (250 μM) for 48 h ($n$ = 3/group). **H**, **I** Expression levels of p-eIF4E, eIF4E, c-Jun, and P-gp (**H**) and relative quantification (**I**) in Caco2 cells transfected with si-NC or si-eIF4E and with or without GDP-L-fucose (250 μM) for 48 h ($n$ = 3/group). Data are presented as mean ± SD. Statistical analyses were conducted using two-way ANOVA (**E**, **G**). Exact $P$ values are reported in the figures.

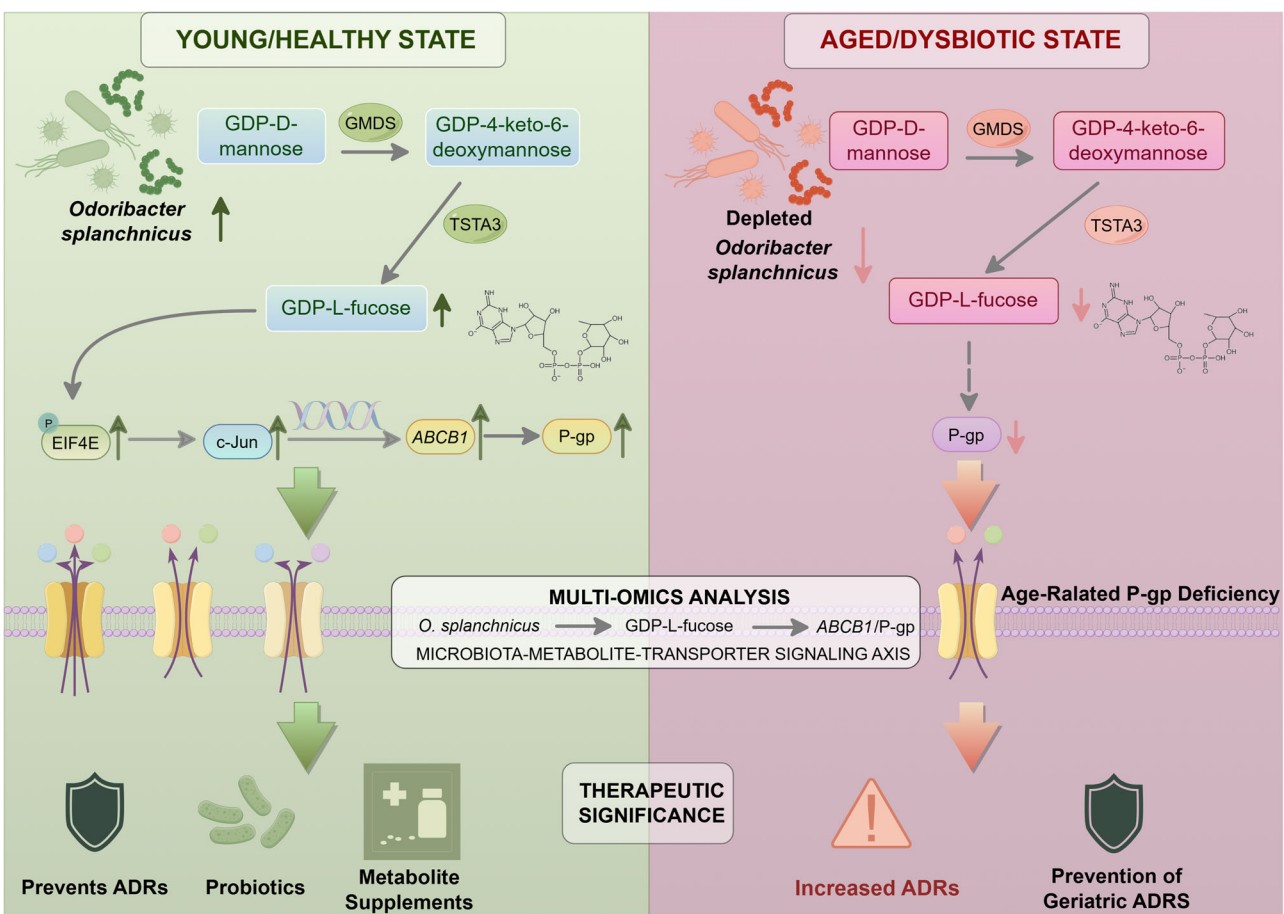

**Fig. 8 | Proposed mechanism of microbiota-mediated regulation of intestinal P-gp.** Aging induces gut microbiota dysbiosis characterized by depletion of *Odoribacter splanchnicus* (*O. splanchnicus*), leading to reduced intestinal P-gp (*ABCB1*) expression and impaired xenobiotic efflux. *O. splanchnicus* encodes GDP-mannose 4, 6-dehydratase (GMDS) and GDP-ʟ-fucose synthase (TSTA3), which mediate microbial biosynthesis of GDP-ʟ-fucose. GDP-ʟ-fucose acts on intestinal epithelial cells to promote phosphorylation of eukaryotic translation initiation factor 4E (eIF4E), thereby activating c-Jun–driven transcription of *ABCB1* and enhancing P-gp expression. This microbiota–metabolite–transporter signaling axis preserves intestinal detoxification capacity, and its disruption contributes to age-associated decline in P-gp function. Created by figdraw.com (Copyright Code: ARORA33111).

---

(PM0640-250g, Baigen) supplemented with 0.5% L-cysteine (AA1048, HARVEYBIO) and 0.5% *N*-acetyl-ᴅ-glucosamine (M1001-100g, MREDA) at 37 °C in an anaerobic chamber (5% $H_2$, 10% $CO_2$, 85% $N_2$). Resazurin Azurin (R1090-5g, Biotopped) was added as an oxygen indicator. Cells were stored as anaerobically prepared 25% (*v/v*) glycerol stocks in sealed anaerobic glass tubes for long-term storage. All manipulations with *O. splanchnicus* were performed within the anaerobic chamber using pre-produced medium and reagents.

After 16–24 h, *O. splanchnicus* reached the logarithmic growth stage, and the bacterial fluid became cloudy. The light absorption value of the bacterial solution at 600 nm was measured, and bacterial counts were determined. Following centrifugation at $10,000 \times g$ for 20 min, a $1 \times 10^9$ CFU/mL bacterial suspension was prepared using PBS.

Three-month-old or 22–26-month-old mice were treated with Abx for 2 weeks, then randomly assigned to the experimental group. Each mouse received 200 μL of *O. splanchnicus* suspension via gavage once daily for 2 weeks. Intestinal tissues from the experimental group were collected for follow-up analysis.

**Construction and transplantation of engineered bacteria**

In this study, we aimed to construct and transplant engineered bacteria. Two genes: *Gmd* (Gene ID: 946562) and *WcaG* (Gene ID: 946563), were cloned from the *Escherichia coli* MG1655 genome (sizes: 1122 bp and 963 bp, respectively) into the plasmid vector

pETDuet-1. The *Gmd* gene was inserted at the XbaI restriction enzyme site, and the *WcaG* gene at the NdeI site. The constructed plasmids were transformed into *E. coli* DH5α competent cells, and positive transformants were selected for DNA sequencing to confirm successful gene insertion. Ultimately, we obtained the recombinant plasmid pET-*Gmd*-*WcaG*, which was further transformed into *E. coli* B strain BL21 (DE3) to generate the engineered bacteria EGW. This entire process was performed by Shanghai Newpu Biotechnology Co., LTD.

To culture the engineered bacteria, single colonies were inoculated into LB liquid medium supplemented with Ampicillin (A800200-25g, Macklin, 100 μg/mL) and incubated overnight. Subsequently, a 0.5 mL aliquot of the culture was transferred to a 250 mL shake flask containing 25 mL of LB medium with ampicillin, and incubated at 37 °C with shaking at 200 rpm. The culture was incubated at 37 °C with shaking at 200 rpm. When the bacterial OD600 reached a range of 0.6–0.8, we induced protein expression using IPTG (final concentration: 0.5 mmol/L) and lowered the temperature to 25 °C for continued cultivation.

After approximately 20 h of fermentation in shake flasks, fermentation broth samples were collected and processed. Wild-type *E. coli* (EC) was cultured in parallel as a control. After centrifugation at $10,000 \times g$ for 20 min, a bacterial suspension ($1 \times 10^9$ CFU/mL) was prepared using PBS. Mice in all groups were administered Abx for

2 weeks, followed by daily gavage of 200 μL of the bacterial suspension for another 2 weeks. Intestinal tissues, feces, and small intestine contents were collected from the experimental group for follow-up detection.

### GDP-L-fucose treatment

Three-month-old mice in the experimental group were randomly divided into two groups and treated with Abx for 2 weeks. After that, mice were provided with drinking water containing 150 mg/L GDP-L-fucose disodium salt (G409416, Aladdin), which was changed every other day for 2 weeks. Intestinal tissues from the mice in the experimental group were collected for follow-up detection.

### Bacteria colonization PCR

Fecal bacterial genomic DNA was extracted from bacteria-transferred mice using the TIANamp Bacteria DNA Kit (DP302, TIANGEN) and TIANamp Stool DNA Kit (DP328, TIANGEN). PCR was performed with 2× EasyTaq® PCR SuperMix (+dye) (AS111, TransGen Biotech), followed by electrophoresis on a 2% agarose gel. Real-time qPCR was conducted using TransStart® Top Green qPCR SuperMix (+Dye II) (AQ132-24, TransGen Biotech). The relative abundances of specific taxa were normalized with universal bacterial primers.

### Paraffin section and immunohistochemical staining of intestinal tissue

Fresh intestinal tissues were washed with PBS buffer and fixed in 4% paraformaldehyde at 4 °C for 48 h, then dehydrated in 20% sucrose solution for 24 h. Subsequently, the intestinal tissues were embedded in paraffin, and sectioned at a thickness of 7 μm. Sections were dewaxed and rehydrated using xylene and gradient ethanol. Immunohistochemical staining of intestinal cross-sectional sections was performed using a two-step universal reagent kit (mouse/rabbit high-sensitivity polymer detection system) (PV-8000, ZSGB Bio) and an anti-P glycoprotein antibody (ab170904, Abcam, 1:200). IHC images were captured by the Grundium Ocus® and processed by Aperio Image Scope 12.4.6 (Leica Microsystems IR GmbH) and Image J software (Wayne Rasband).

### FW preparation

Frozen fecal samples were used to prepare FW. Briefly, feces were diluted in PBS at a ratio of 400 mg feces to 1 mL PBS, homogenized using a Tissue Lyser II (Qiagen, Germany) at 30 Hz for 4 min, and centrifuged at 10,000×g for 10 min at 4 °C. The mixture was homogenized for 4 min at 30 Hz using a Tissue Lyser II (Qiagen, Germany). The homogenate was then centrifuged at 10,000×g for 10 min at 4 °C. The resulting supernatant was further centrifuged at 2000×g for 3 min at 4 °C. The supernatant was further centrifuged at 2000×g for 3 min at 4 °C, then sterile-filtered to obtain FW for subsequent analysis.

### Cell culture and treatment

Caco2 cells, purchased from the American Type Culture Collection (ATCC), were cultured in Caco2 cell culture medium (CM-0050, Procell) and maintained in a humidified atmosphere with 5% $CO_2$ at 37 °C. LS180 and T84 cells were cultured at 37 °C in DMEM supplemented with 10% FBS, 1% nonessential amino acids, and 1% penicillin-streptomycin, also maintained in a humidified atmosphere with 5% $CO_2$ at 37 °C[48,49].

Caco2, LS180, and T84 cells were seeded in 6-well plates at a density of $1 \times 10^6$ cells/well. After 48 h of growth, the cells were treated with GDP-L-fucose disodium salt (G409416, Aladdin) at 250 μM for 48 h or FW at 200 μL/well for 48 h. Subsequently, the cells were washed and collected for protein extraction and RNA extraction, followed by gene expression analysis.

### RNA-seq analysis

Caco2 cells were seeded in 6-well plates, with five samples treated with 250 μM GDP-L-fucose, five treated with 200 μL/well *O. splanchnicus* supernatant, and five as negative controls. Specifically, five samples were treated with GDP-L-fucose (250 μM), five samples were treated with *O. splanchnicus* supernatant (200 μL/well), and five negative control samples were included. Total RNAs were isolated using the TRIzol reagent, and RNA-seq analysis was conducted by BGI Genomics Co., Ltd.

### Transcription factor prediction

The promoter sequence of the *ABCB1* gene, located on chromosome 7 in the GRCh38.p14 assembly, was identified using the UCSC Genome Browser. The GRCh38.p14 assembly is a reference genome version for Homo sapiens. The PROMO usage database was employed to analyze the *ABCB1* promoter sequence (http://alggen.lsi.upc.es/cgi-bin/promo_v3/promo/promoinit.cgi?dirDB=TF_8.3). Integrating literature and RNA-seq data, it is predicted that *ABCB1* may be regulated by approximately 40 transcription factors.

### ChIP-PCR, ChIP-qPCR

Caco2 cells were cultured in a 10 cm cell culture dish with 10 mL of medium. When the cell density reached 80–90%, formaldehyde was added directly to the medium to achieve a final concentration of 1%. The mixture was then incubated at 37 °C for 10 min to cross-link the target protein and its corresponding genomic DNA. Enzymatic ChIP was performed using the following components: Protein A/G Magnetic Beads (P2083S, Beyotime), c-Jun (60A8) Rabbit Monoclonal Antibody (9165S, Cell Signaling Technology), Histone H3 Rabbit Polyclonal Antibody (AF7101, Beyotime) (used as a positive control), Rabbit IgG (A7016, Beyotime) (used as a negative control). ChIP-DNA samples were purified using a PCR/DNA purification kit (D0033, Beyotime). The obtained ChIP-DNA was subsequently analyzed by PCR and qPCR (Table S3). Finally, the PCR products were separated by electrophoresis on a 2% agarose gel.

### siRNA and plasmid transfection

Caco2 cells were seeded in 6-well plates. The cells were then transfected with either specific siRNA or a control siRNA (provided by GenePharma, Shanghai, China).

Caco-2 cells were seeded in 6-well plates and subsequently transfected with either the GFP-*EIF4E* or GFP-*HPRT1* plasmids (provided by Tsingke Biotechnology Co., Ltd., Beijing, China).

The transfection was carried out using the jetPRIME transfection reagent (101000046, Polyplus) for 48 h. The sense strand sequences for the specific siRNAs targeting the *JUN* and *EIF4E* gene were as shown in Table S4. And the plasmid sequences of GFP-*EIF4E* and GFP-*HPRT1* are shown in Table S5. After transfection, cellular proteins and mRNA were extracted for further analysis.

### Real-time qPCR

Total RNA was extracted using TRIzol reagent (15596018CN, Invitrogen) and reverse-transcribed into cDNA using the TransScript® One-Step gDNA Removal and cDNA Synthesis SuperMix kit (AT311-03, TransGen Biotech) according to the manufacturer's protocol. mRNA levels were normalized to the housekeeping gene β-actin, and relative expression levels were calculated using the $2^{-\Delta\Delta Ct}$ method.

### Western blotting

Cellular and tissue proteins were dissociated using RIPA lysis buffer (C1053-100, APPLYGEN). This step helps to extract proteins from the cells and tissues. The following antibodies were used to label specific protein bands:

GAPDH monoclonal antibody (60004-1-Ig, Proteintech, 1:3000)
Anti-P Glycoprotein antibody (ab170904, Abcam, 1:1000)
GMDS polyclonal antibody (15442-1-AP, Proteintech, 1:500)
TSTA3 polyclonal antibody (15335-1-AP, Proteintech, 1:500)
c-Jun (60A8) rabbit mAb (9165S, Cell Signaling Technology, 1:1000)
eIF4E Monoclonal antibody (66655-1-Ig, Proteintech, 1:5000)
Rabbit monoclonal [EP2151Y] to eIF4E (phospho S209) (ab76256, Abcam, 1:2000)
ProteinFind® Anti-GFP Mouse Monoclonal Antibody (HT801-02, TransGen Biotech, 1:2000)
Horseradish peroxidase-conjugated secondary antibodies were used to bind to the primary antibodies. A chemiluminescence detection system was employed to acquire signals from the labeled protein bands.

### Binding prediction
AutoDock 4.2 was utilized to conduct ligand extraction and other operations on the existing structural data obtained through X-ray crystallography, electron microscopy, or Artificial intelligence-predicted models. Docking simulations were performed using GDP-L-fucose (PubChem CID = 135412609), and results were visualized using PyMOL[50,51].

### MST
The MST assay was conducted as previously described[52]. Specifically, the equilibrium dissociation constant (Kd) values were measured using the NanoTemper Monolith NT.115 instrument. GDP-L-fucose was serially diluted to concentrations ranging from 10 nM to 500 mM and incubated with Caco2 cell lysates transfected with GFP-labeled *EIF4E* or *HPRT1* plasmids for 30 min at 25 °C in the dark. The samples were subsequently loaded into glass capillaries (NanoTemper Technologies), and the MST assay was performed following the manufacturer's protocol.

### Statistics and reproducibility
The data represent mean values ± standard deviation (SD) from three or more independent experiments made in three technical replicates. Statistical analyses were performed using GraphPad Prism 8.0. For box plots, the center line represents the median, the box bounds indicate the 25th and 75th percentiles, and whiskers denote the minimum and maximum values. Data normality was assessed using the Shapiro–Wilk test. For comparisons between two groups, normally distributed data were analyzed using a two-sided unpaired *t*-test, and non-normally distributed data were analyzed using the Mann–Whitney *U*-test. For comparisons involving more than two groups, one-way ANOVA (for a single independent factor) or two-way ANOVA (for two independent factors) was used. For correlation analyses, data normality was also assessed using the Shapiro–Wilk test; Pearson's correlation was applied to normally distributed data, while Spearman's rank correlation was used for non-normally distributed data. All statistical tests were two-sided, and exact *P* values are provided in the figure unless otherwise stated. A value of *P* < 0.05 was considered statistically significant. All statistical methods used are indicated in the corresponding figure legends.

### Reporting summary
Further information on research design is available in the Nature Portfolio Reporting Summary linked to this article.

## Data availability
All data supporting the findings of this study are publicly available and have been deposited in appropriate repositories. The microbiome, metabolomics, and RNA-seq datasets generated in this study have been deposited in the OMIX database at the China National Center for Bioinformation/Beijing Institute of Genomics, Chinese Academy of Sciences under accession codes OMIX012398 (human microbiome), OMIX012399 (mouse microbiome), OMIX012397 (mouse intestinal metabolomics), and OMIX012395 (RNA-seq data), accessible at https://ngdc.cncb.ac.cn/omix/. Supporting processed data are also available on figshare under https://doi.org/10.6084/m9.figshare.26801872. All additional data supporting the findings of this study are included within the article and its Supplementary Information File. Transcriptomic data were obtained from ref. 20. and are publicly available at https://www.gutcellatlas.org/spacetime/epithelium/), in compliance with the CC BY 4.0 license terms specified by the original authors.

## Code availability
Our study does not involve the use of any custom-developed code or software. All software used is either open-source (with version-specific documentation and download links provided in the Methods) or commercially available.

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

## Acknowledgements

This work was supported by the National Natural Science Foundation of China [82104293 to Cheng C.]; the Bill and Melinda Gates Foundation [INV-007625 to Dongyang L.]; the National High-tech Research and Development Program [2020YFA0803700; 2023YFA1800904 to Lemin Z.]; and the Natural Science Foundation of Beijing Municipality [L232031; J230039 to Lemin Z.]. The schematic illustrations in this article were either created directly using FigDraw (License Codes: ARORA33111) or prepared in Adobe Illustrator, incorporating licensed graphical elements from FIGDRAW (License Codes: ARSUAb7778) and BioRender (Agreement number: AI28VMQNHZ), with additional original modifications.

## Author contributions

D.L. conceived the project. L.Z. and D.L. designed the experiments. C.C., L.F., and L.L. performed most of the experiments and analyzed the data. X.L., R.Z., Q.Z., R.M., G.H., M.Z., X.W., H.C., F.Z., and H.L. performed part of the experiments. J.E.V.S., J.C., and Z.L. performed the modeling and bioinformatic analyses in this study. C.C. and L.F., and all other authors discussed the data and wrote the manuscript. L.Z. and D.L. provided critical guidance for the revision of the manuscript.

## Competing interests

The author declared no competing interests
