## [Transparent Peer Review file · Nature Communications]

Odoribacter splanchnicus rescues aging-related intestinal P-glycoprotein damage via GDP-L-fucose secretion

Corresponding Author: Professor Lemin Zheng

Version 1:

Reviewer comments:

Reviewer #1

(Remarks to the Author)

The study by Cui et al explore the role of the species *Odoribacter splanchnicus* in rescuing age-related intestinal P-Glycoprotein damage via GDP-L-Fucose secretion. The study is interesting and provides further experimental evidence on the anti-aging role of *Odoribacter*. However, I do not concur with the authors' view that *Odoribacter* is only a youth-associated species. It is a species also associated with healthy aging individuals including centenarians (Please see: Ghosh et al *Nature Reviews Gastroenterology and Hepatology*: <https://doi.org/10.1038/s41575-022-00605-x>; Ghosh et al *Nature Aging*: <https://doi.org/10.1038/s43587-022-00306-9>; Sato et al *Nature*: <https://doi.org/10.1038/s41586-021-03832-5>).

While I do appreciate the detailed and extensive investigations to show the influence of the gut microbiota on the P-gp expression, including the identification of GDP-L-Fucose and the identification of the enzymes GMDS and TSTA3. I did not see any comprehensive results section that shows this in a comprehensive manner. Metabolomic data is cryptic table of p-values without annotation of any metabolite and their taxonomic comparison table is highly confusing. I see other microbes belonging to Bacteroidetes phylum besides *Odoribacter* here as well. Infact except the figures, most of the results are too cryptic to trust.

The authors also have the same cryptic way of writing the manuscript when it comes to describing their methods. Some of the examples are below:

a. Figure 1A-1B: The authors do not provide details of the meta-analysis that they claim to have performed. Though they mention screening of clinical trials etc, in the Methods section. There is no mention of the number of studies, the effect size computation, which models were used. They simply mention using 'Reveiw Manager 5.3 software', which clearly shows that they themselves are not aware of the methodological details. This is not acceptable.

b. Same goes for 'Two additional meta-analyses' for S1A.

c. "Human colon adenocarcinoma (Caco2) cells, when grown on semipermeable filters, spontaneously differentiate in culture to form confluent monolayers which both structurally and functionally resemble the small intestinal epithelium". Do they have any evidence in literature?

4. The manuscript has terrible English and the language needs drastic refinement.

5. Lastly, I still do not understand why *Odoribacter*. Although I agree that it shows consistent change in both mice and human samples, there are other *Bacteroides* as well. The sample sizes of the mice and the human cohorts here are too small. Unless they are able to replicate their three-way findings on GMDS and TSTA3 enzyme abundance, *Odoribacter splanchnicus* and aging in other larger human cohorts or large mice cohorts, I cannot recommend this manuscript for publication in *Nature Communications*.

5. While the authors have performed extensive investigations

Reviewer #2

(Remarks to the Author)

In their work, Cui et al. propose aging-related changes in the abundance of a particular bacterial species, *Odoribacter splanchnicus*, as a causative agent in aging-associated suppression of P-glycoprotein (P-gp). P-gp in turn is an important transporter involved in xenobiotic efflux in the small intestine and its reduced expression with age is associated with an increased likelihood of adverse drug reactions due to less efficient detoxification/disposal of pharmaceuticals. On the host side, they show that L-fucose is involved in a signaling cascade that induces P-gp expression in epithelial cells. I find this work elegantly addresses a quite important aspect of microbiome-host-interactions in aging. However, I do think that some of the aspects described in the work are not fully supported by the data.

Major points

1) The authors identify *O. splanchnicus* as a main producer of L-fucose in both humans and mice and argue that the reduced abundance of this species is causally involved in aging-associated decline of P-gp. However, this mechanistic connection is only poorly supported by the data. They observe both in humans as well as in mice, a large number of aging-regulated species. Particularly, they observe a suppression of *Bacteroides* species. On the other hand, they find that *Bacteroides* are a major clade containing the enzymes for L-fucose production from GDP-D-mannose. Moreover, they find the relevant enzymes also in many other species present in the gut such as *Bacillota* or *Pseudomonadota*. Even if those enzymes seem to be enriched in *Bacteroides*, this does not necessarily prove that the demise of this particular species actually mediates aging-associated decline of L-fucose production. They do show that the aging microbiome modulates P-gp expression and that L-fucose exerts an effect on P-gp expression, but the data does not support that it is specifically L-fucose production by *O. splanchnicus* that mediates the effect of the aging microbiome on P-gp expression.

2) L-fucose is also an important component of mucins that is released by mucin-degrading bacteria. Since there is also an aging-associated decline in mucin expression, can the authors exclude that the effect of fecal water from aged participants on P-gp expression is not primarily driven by reduced L-fucose release due to reduced mucin availability in the old?

3) Can the authors discuss a bit more about the origins of GDP-D-mannose that is the precursor of L-fucose. Is it produced/released by host cells or synthesized de novo by bacteria?

4) The authors state that the major site of action of L-fucose mediated suppression of P-gp is the small intestine. However, microbial abundance in this part of the gastrointestinal tract is way lower than in the colon. It should be discussed which role colonic expression of P-gp plays in the observed increased likelihood of adverse drug reactions due to P-gp suppression in age.

5) Also along these lines, most of the microbiome data the authors generate are from fecal samples and thus from colon. Since microbiota from colon and small intestine differ and *Bacteroides* levels in small intestine are typically smaller than in the colon, can they actually show that *O. splanchnicus* is actually present in the small intestine and declining with age? I guess corresponding data for humans might be hard to come by, but at least in mice this should be possible to determine.

Minor:

- Fig. 1E - doesn't really contain information and can be removed
- Fig. 1F-I: font sizes in the figure are too small to be readable, particularly in Fig. 1I
- Fig. 3H and I not readable

Reviewer #3

(Remarks to the Author)

Chen Cui et al report on *Odoribacter splanchnicus* which rescues aging-related intestinal P-glycoprotein damage via GDP-L-Fucose secretion.

Specifically, they propose gut microbiota dysbiosis as a key driver of age-associated P-gp deficiency.

They performed multi-omics analyses of human cohorts and murine models. They find that *Odoribacter splanchnicus* (*O. splanchnicus*) has a superior enzymatic activity and capacity to enzymatically provide the relevant biosynthesis of the microbial derived metabolite GDP-L-Fucose. This in turn should then result in GDP-L-Fucose activating c-Jun-mediated transcriptional programming through phosphorylation-dependent enhancement of eukaryotic translation initiation factor 4E (eIF4E), thereby upregulating ABCB1 expression and - most importantly, restoring P-gp-mediated xenobiotic clearance.

Aged humans and senescent mice display parallel depletion of colonic *O. splanchnicus* and GDP-L-Fucose levels appeared to display parallel depletion of colonic *O. splanchnicus* and GDP-L-Fucose levels, that are considered as causative factors in P-gp dysfunction. The authors therefore postulate a microbiota-metabolite signaling axis (*O. splanchnicus* → GDP-L-Fucose → eIF4E/c-Jun → ABCB1).

As a consequence, the authors propose and hope to prevent adverse drug reactions in the elderly by microbiome targeted interventions.

The question is important and microbiome targeted interventions are timely, particularly in the elderly with a changing composition of the gut microbiome and the risk of ADR. A potential reduction of ADR by "correction" of the gut microbiome appears therefore attractive. The experiments appear to have been performed carefully and are described in detail. Mechanistic approaches by specific transplants and ex vivo as well as in vivo data are logically provided.

The following points should be considered:

1. Intro: The role of P-GP is overfocussed and overemphasized in the elderly, sind eg for DOACs many other mechanisms contribute to different plsama concentrations including liver insufficiency, cardiac function, altered kidney function polymedication and medication interactions etc etc. This is later enumerated and mentioned that it was corrected for in earlier work, but this correction may be not trivial given the continuum of organ dysfunction.
2. Particularly bleeding complications should be attributed carefully, given the many concomitant aspects involved (including eg dual anticoagulation and other medication interactions.eg line 127.
- 3.Likewise, elderly people have per se higher bleeding risks, even in the absence of concomitant P-GP interfering drugs.
- 4.The n number in fig 1 seems very small.
5. The take home fig is oversimplified and needs revision
6. Explain a potential future targeted intervention in the elderly either by bacterial transplation with or without antibiotic treatment and or by providingL-Fucose.
7. Enthusiasm about the application of odoribacter splanchnicus must be tempered since it has been shown to associate with several serious diseases, incl intracranial, cardiovasc, kidney - please consider cite and discuss, eg Janhong Li in recent *Microorganisms* 2025, 13(4), 815; <https://doi.org/10.3390/microorganisms13040815>.
8. Under specific pathological conditions, such as exacerbated intra-abdominal inflammation or impaired intestinal barrier function, it may act as an opportunistic pathogen, triggering infections or even sepsis. Please discuss.
- 9.Specificity: Another mechanism of protection has been described of *D. splanchnicus* by inhibition of other pathogens leke *Salmonella*, as reported in doi: <https://doi.org/10.1101/2024.08.23.609322>, via its secreted bacteriocin.
10. Short chain fatty acids are produced by certain gut bacteria, also by *D splanchnicus*? Were they also restoring the P-GP activity?

Version 2:

Reviewer comments:

Reviewer #1

(Remarks to the Author)

While I appreciate the efforts of the authors in refining the version 1 of this manuscript, some of my previous concerns remain.

1. I believe that O.S is associated with a reproducible age-related decline, but I am bit skeptical about the justifications provided by the authors regarding the other functional links. GMDS and TSTA3 are encoded by other members of the *Bacteroides* phylum, many of which show the trend in both human and mice (decline in old). The authors show that supplementation by O.S increases expression of these enzymes, but what about other *Bacteroides* members. Does supplementation by other *Bacteroides* members not show these trends? There are multiple such members which show decline in age? What about *Alistipes*, *Parabacteroides*, *Barnesiella*, etc? The authors need to clearly show that supplementation by any of these other members does not show these changes.

2. I want to add a minor correction to the responses of the authors. The results shown by the authors in page 9 of their responses pertains to the paper by Wang et al in *Nature Medicine* (<https://www.nature.com/articles/s41591-024-03038-y>) not Ghosh et al (which is a different study in *Nature Aging* also highlighting O.S). However, my response to the clarifications/observations of the authors in this specific context is this. The Supplementary table highlighted by the authors also contains the multiple *Bacteroidetes* phylum members, some of which show even stronger negative associations with age (especially in the MC1 or normal metabolic cluster). Hence, why *Odoribacter*?

To summarize, I am not saying that the authors' O.S (*Odoribacter splanchnicus*) is erroneous. I just want the authors to highlight that there are other members within the same clade as well who show this property (or if not provide validation experimental results to show that it is not so).

And provide a much comprehensive data-driven investigation showing the presence of GMDS and TSTA3 homologs across all clades of human gut microbes and show that O.S and its relatives are the only ones that harbor these.

Reviewer #2

(Remarks to the Author)

I thank the authors for thoroughly addressing all of my concerns.

Reviewer #3

(Remarks to the Author)

The authors have responded adequately to most of the Reviewers questions.

Where some doubts of the magnitude of the effects and their specificity remain, the authors have now stated this in the ms in the discussion and where appropriate.

The authors have performed additional experiments and literature review to substantiate their findings and to respond to the Reviewers points.

REVIEWER COMMENTS

Reviewer #1 (Remarks to the Author):

The study by Cui et al explore the role of the species *Odoribacter splanchnicus* in rescuing age-related intestinal P-Glycoprotein damage via GDP-L-Fucose secretion. The study is interesting and provides further experimental evidence on the anti-aging role of *Odoribacter*.

Response to Reviewer #1:

We sincerely thank you for the thoughtful and positive evaluation of our work. We are pleased that you found the study interesting and recognized its contribution in providing experimental evidence for the role of *O. splanchnicus* in counteracting age-related intestinal P-glycoprotein decline through GDP-L-fucose secretion. In the following sections, we address each of your comments point by point, providing clarifications, additional data where relevant, and revisions to the manuscript.

However, I do not concur with the authors' view that *Odoribacter* is only a youth-associated species. It is a species also associated with healthy aging individuals including centenarians (Please see: Ghosh et al Nature Reviews Gastroenterology and Hepatology: <https://doi.org/10.1038/s41575-022-00605-x>; Ghosh et al Nature Aging: <https://doi.org/10.1038/s43587-022-00306-9>; Sato et al Nature: <https://doi.org/10.1038/s41586-021-03832-5>).

Response:

We sincerely thank you for this insightful comment and for pointing us to the relevant and important literature (Ghosh et al., Nat Rev Gastroenterol Hepatol, 2022; Ghosh et al., Nat Aging, 2022; Sato et al., Nature, 2021). We fully agree with you that *O. splanchnicus* is not exclusively a youth-associated species. Indeed, several studies, including those cited by you, demonstrate that *O. splanchnicus* can also be enriched in healthy elderly populations, particularly in centenarians, where it has been linked to resilience against age-related decline. Our study emphasizes that during typical aging, *O. splanchnicus* and its metabolite GDP-L-fucose decline in abundance, contributing to impaired intestinal P-glycoprotein activity. The enrichment observed in long-lived cohorts may thus represent a protective microbial signature of healthy aging.

To address this important point, we have revised the manuscript to clarify our interpretation, explicitly acknowledging prior reports of *O. splanchnicus* enrichment in centenarians and healthy aging cohorts, and discuss how our findings complement these observations by highlighting its protective role.

Updated Text for Discussion (Page 12, Line 279~283 to Page 13, Line 284~286 in the revised manuscript):

“*O. splanchnicus* is a beneficial gut commensal whose abundance declines with age and deteriorating health, a trend we observed in our study and that was independently validated in a large cohort of 10,207 individuals aged 40–93 years with follow-up data³². Notably, it remains enriched in healthy older adults and centenarians, suggesting a

potential role in resilience against age-related physiological decline³³⁻³⁷. Its abundance is associated with favorable metabolic and intestinal profiles, and depletion has been linked to inflammatory and metabolic disorders, including Inflammatory Bowel Disease (IBD), fatty liver disease, and colon cancer^{38,39}.”

While I do appreciate the detailed and extensive investigations to show the influence of the gut microbiota on the P-gp expression, including the identification of GDP-L-Fucose and the identification of the enzymes GMDS and TSTA3. I did not see any comprehensive results section that shows this in a comprehensive manner. Metabolomic data is cryptic table of p-values without annotation of any metabolite and their taxonomic comparison table is highly confusing. I see other microbes belonging to Bacteroidetes phylum besides *Odoribacter* here as well. Infact except the figures, most of the results are too cryptic to trust.

Response:

We sincerely thank you for this critical comment, which we feel has greatly improved the clarity of our Results section. We apologize that our original presentation appeared cryptic. In the revised manuscript, we have made several important clarifications and additions.

Selection of microbial candidates: In our multi-omics analyses, we prioritized taxa that exhibited consistent and significant alterations in both human cohorts and murine models. This stringent cross-species filtering led us to focus on *O. splanchnicus*, which not only showed marked depletion with aging but also demonstrated the strongest functional link with intestinal P-gp expression. We have clarified this rationale in the Results section.

Metabolomic data presentation: In the revised version, we have systematically annotated each metabolite in the uploaded dataset “Data_Untargeted metabolomics analysis of small intestine tissue in mus-y and mus-o groups in Fig4” (available in the Figshare repository). For each metabolite, we provide its name, log₂ (fold change), p-value, and KEGG/HMDB/PubChem/ChEBI identifiers where available, with the list arranged by |log₂FC| for clarity. Pathway ID and Name are also included to facilitate functional interpretation. To further aid understanding, we complemented the dataset with visualization in the revised manuscript: a volcano plot (Fig. 4H) highlights the distribution of significant metabolites, and KEGG pathway enrichment analysis (Fig. 4I) places these differences into the context of relevant metabolic pathways. For transparency and accessibility, the complete raw metabolomic dataset has been deposited in the Figshare database under the above file name and is also provided as a supplementary attachment to this submission. Together, these improvements make the metabolomic results more interpretable and directly linked to biological functions.

Functional validation: To provide causal evidence beyond correlative analyses, we performed a supplementation experiment in naturally aged mice. Specifically, 22–24 month old C57BL/6 mice were pretreated with antibiotics (Abx) for 2 weeks and then gavaged with either PBS control (Abx-o-Con) or *O. splanchnicus* (Abx-o-OS) for 2 weeks (n=8/group). Successful colonization of *O. splanchnicus* was confirmed by qPCR of bacterial DNA in intestinal contents. Importantly, *O. splanchnicus* gavage significantly elevated P-gp expression in both the jejunum and ileum (Fig. S5 in the

revised manuscript). These results provide direct functional validation that *O. splanchnicus* colonization is sufficient to restore intestinal P-gp expression in aged hosts.

Correlation analyses: We now provide detailed correlation analyses (Fig. S7 in revised manuscript) showing that *O. splanchnicus* expresses the key enzymes GMDS and TSTA3, produces GDP-L-fucose as confirmed by LC-MS, and that its abundance positively correlates with P-gp (*ABCB1*) expression in both jejunum and ileum of young and aged mice (n=20/group). Moreover, *O. splanchnicus* abundance also correlates with DNA levels of *Gmd* and *WcaG*, as well as functional orthologs (COG1089, K01711, K02377) detected in metagenomic datasets, further supporting its enzymatic role in GDP-L-fucose biosynthesis.

Updated Text for Results (Page 7, Line 150~163 to Page 9, Line 188~191 in the revised manuscript):

“P-gp Upregulation by *O. splanchnicus* and GDP-L-fucose

To identify specific gut microbes responsible for regulating intestinal P-gp, we profiled human fecal metagenomes (10 young adults, HY; 9 elderly, HO, Table S2) and murine 16S rRNA sequencing (10 juvenile, mus-y; 10 aged, mus-o). Both analyses revealed marked restructuring of microbial communities with age, as reflected by beta-diversity analyses at family, order, and genus levels (Figs. 3A–C, 3E–F, and S3a–d). This decline was accompanied by consistent depletion of *Odoribacter*, particularly *O. splanchnicus*, in both species (Figs. 3D–L and S3e–f). Independent validation using PCR and qPCR confirmed a significant reduction of *O. splanchnicus* in the feces and small intestinal contents of aged mice (Fig. S3g–l). Cross-species prioritization thus highlighted *O. splanchnicus* as the most robust age-depleted taxon potentially linked to intestinal P-gp expression.

To evaluate its functional role, 3-month-old C57BL/6 mice were pretreated with antibiotics and subsequently colonized with either PBS (Abx-Control) or *O. splanchnicus* (Abx-OS) (Figs. 4A and S4a–d). Abx-OS mice exhibited significantly increased P-gp protein and *Abcb1* mRNA levels in jejunum and ileum, confirmed by Western blot, qPCR, and IHC (Figs. 4B–G and S4e–h). Similar restoration was observed in aged, antibiotic-pretreated mice gavaged with *O. splanchnicus* (Fig. S5a–f).

Untargeted metabolomic profiling of murine small intestine demonstrated that the relative abundance of GDP-L-fucose was markedly higher in young mice than in aged mice ($\log_2FC = 5.38$), whereas other microbial metabolites, including short-chain fatty acids and related derivatives, declined modestly (Fig. 4H). KEGG pathway enrichment consistently highlighted fructose/mannose metabolism as the dominant age-associated pathway across human metagenomes, murine 16S data, and intestinal metabolomics (Figs. 4I–J and S6a–b). Key enzymes in this pathway, GDP-mannose 4,6-dehydratase (GMDS, COG1089, K01711) and GDP-L-fucose synthase (TSTA3, K02377), were significantly enriched in young groups (Fig. 4K–L).

Survey of 1,647 Human Microbiome Project genomes revealed that GMDS and TSTA3 are prevalent in *Bacteroidota*, including *O. splanchnicus* (Fig. S6c). Western blot and qPCR confirmed higher GMDS/TSTA3 expression in feces and small intestine contents of young versus aged mice (Fig. S6d–h). Colonization with *O. splanchnicus* increased GMDS/TSTA3 levels (Fig. S6i–n), and both enzymes were detectable in *O.*

splanchnicus itself (Fig. S7a). Colonization with *O. splanchnicus* promoted GDP-L-fucose production, verified by LC–MS (Fig. S7b–c). *O. splanchnicus* abundance correlated positively with *Gmd/WcaG* levels, supporting an *O. splanchnicus* – GMDS/TSTA3–P-gp regulatory axis (Fig. S7d–i).

Functional validation demonstrated that exogenous GDP-L-fucose (250 μ M, 48 h) significantly upregulated P-gp expression in Caco2, LS180 and T84 cells (Figs. 4M–N and S8a–d). In vivo, antibiotic-pretreated mice gavaged with GDP-L-fucose (150 mg/L, 2 weeks) exhibited increased P-gp and *Abcb1* expression in jejunum and ileum (Figs. 4O–U, S8e–h).

Overall, these findings indicate that *O. splanchnicus* regulates intestinal P-gp expression via the GMDS/TSTA3-GDP-L-fucose axis. Age-related decline in *O. splanchnicus* and the associated biosynthetic pathway likely contributes to diminished P-gp function in older adults, providing mechanistic insight into microbiota-mediated regulation of intestinal drug transport.”

We believe these revisions significantly strengthen the coherence of our Results section, making the data more comprehensive and easier to follow. We sincerely appreciate your rigorous evaluation and valuable feedback on improving the clarity and depth of our results presentation.

Figure S5
Gavage of *O. splanchnicus* to aged mice treated with Abx can elevate the expression levels of P-gp in their small intestines.

a-f. C57BL/6 WT 22-24 month old mice were treated with Abx for 2 weeks, then, treated with PBS control (Abx-o-Con) or *O. splanchnicus* (Abx-o-OS) for 2 weeks

(n=8/group).

- Overview of animal experiments of Abx-o-Con and Abx-o-OS mouse models.
- qPCR with small intestine contents bacterial DNA to validate the successful colonization; *** $p < 0.001$, t test (and nonparametric test).
- The expression levels of P-gp in jejunum of Abx-o-Con and Abx-o-OS groups.
- The relative expression levels of proteins in jejunum were analyzed using Image J software; * $p < 0.05$, t test (and nonparametric test).
- The expression levels of P-gp in ileum of Abx-o-Con and Abx-o-OS groups.
- The relative expression levels of proteins in ileum were analyzed using Image J software; * $p < 0.05$, t test (and nonparametric test).

Figure S7 d-i

***O. splanchnicus* expresses GMDS and TSTA3, metabolizes GDP-L-fucose, and its abundance positively correlates with small intestinal P-gp expression.**

- Correlation between *O. splanchnicus* abundance in small intestinal contents and jejunal P-gp expression of mus-y and mus-o groups (n=20/group).
- Correlation between *O. splanchnicus* abundance in small intestinal contents and ileum P-gp expression of mus-y and mus-o groups (n=20/group).
- Correlation between *O. splanchnicus* abundance in small intestinal contents and *Gmd* DNA levels of mus-y and mus-o groups (n=20/group).
- Correlation between *O. splanchnicus* abundance in small intestinal contents and *WcaG* DNA levels of mus-y and mus-o groups (n=20/group).
- Correlation between *unidentified_g_Odoribacter* abundance in fecal 16S analysis and levels of COG1089, K01711, K02377 of mus-y and mus-o groups (n=10/group).
- Correlation between *uncultured_bacterium_g_Odoribacter* abundance in fecal 16S analysis and levels of COG1089, K01711, K02377 of mus-y and mus-o groups (n=10/group).

Correlation analysis: Normality assessed by Shapiro-Wilk test; Pearson correlation used for normally distributed data, Spearman rank correlation for non-normal data.

The authors also have the same cryptic way of writing the manuscript when it comes to describing their methods. Some of the examples are below:

- Figure 1A-1B: The authors do not provide details of the meta-analysis that they claim to have performed. Though they mention screening of clinical trials etc, in the Methods section. There is no mention of the number of studies, the effect size computation,

which models were used. They simply mention using 'Reveiw Manager 5.3 software', which clearly shows that they themselves are not aware of the methodological details. This is not acceptable.

b. Same goes for 'Two additional meta-analyses' for S1A.

Response:

We sincerely thank you for raising this important point. In the revised manuscript, we have clarified the methodology in detail. The P-gp substrate pharmacokinetic analysis in Fig. 1A and 1B was not a formal meta-analysis but a systematic literature review with pooled descriptive comparison. We have revised the text to accurately describe it as such and clarified that, due to limited and heterogeneous data, no effect size modeling was performed. For the two DOAC analyses (age-related bleeding risk and concomitant P-gp inhibitor effect, Fig. S1A-B), we have now specified the eligibility criteria, number of included trials, and statistical methods.

We believe these revisions address your concerns and improve the transparency of our methodology.

Figure S1c. DOAC bleeding risk and age meta-analysis workflow

Figure S1d. DOAC bleeding risk and P-gp inhibitor meta-analysis workflow

Updated Text for Methods (Page 15 to Page 16, Line 342~353 in the revised manuscript):

“Systematic review of pharmacokinetic data for P-gp substrates

We searched PubMed (up to August 2024) for clinical studies reporting pharmacokinetics of FDA-defined P-glycoprotein (P-gp) substrates (dabigatran etexilate, digoxin, edoxaban, fexofenadine) in young and older adults. Keywords were predefined, and studies were eligible if they stratified pharmacokinetic outcomes (AUC, Cmax) by age and reported sample size, demographics, dosing, and variability. In total, we identified 7 studies on digoxin, 3 on fexofenadine, 6 on dabigatran, and 2 on edoxaban. Owing to the limited number and heterogeneity of such studies, no meta-analysis was performed. Instead, parameters were descriptively summarized and compared across age groups using GraphPad Prism v8.0. The “n” shown in Fig. 1A-B directly reflects the sample sizes of the original studies. Across all available reports, older adults consistently exhibited higher systemic exposure to P-gp substrates, supporting the hypothesis that age-related reductions in P-gp function contribute to altered drug disposition.

Meta-analyses of bleeding risk with DOACs

To evaluate clinical correlates of P-gp function, we conducted two independent meta-analyses of randomized controlled trials (RCTs) involving direct oral anticoagulants (DOACs: dabigatran etexilate, edoxaban, rivaroxaban, apixaban).

Age-related bleeding risk. A comprehensive search of the FDA drug database and ClinicalTrials.gov was performed to identify all published phase III RCTs of FDA-approved DOACs. Exclusion criteria were: (i) ANDA drugs, (ii) non-phase III trials, (iii) patient age <18 years, (iv) non-oral administration, (v) lack of randomization, (vi) absence of age-stratified outcomes, (vii) multi-drug regimens, and (viii) non-targeted drugs. After applying these criteria, 12 eligible trials were included. The primary endpoint was major bleeding, compared between younger and older age groups across different DOAC doses (Fig. S1c).

Impact of concomitant P-gp inhibitors. The same search and screening strategy was applied, with the additional requirement of reporting concomitant P-gp inhibitor use. After exclusions, two RCTs (dabigatran etexilate and edoxaban) were included. Major bleeding events were compared between patients treated with or without P-gp inhibitors under single-agent DOAC therapy (Fig. S1d).

For both analyses, individual study data were pooled using Mantel-Haenszel (M-H) fixed-effect model. Risk Ratio (RR) with its corresponding 95% confidence interval (CI) was calculated as combined effect measure. Statistical heterogeneity of overall or subgroup was calculated using Chi-square test and quantified using I². An I² value larger than 50% revealed that there was a substantial heterogeneity. The statistical significance of overall effect was assessed by Z-test, with a P-value less than 0.05 considered statistically significant. All analyses were conducted using Review Manager (RevMan, v5.3, Cochrane Collaboration, UK).”

"Human colon adenocarcinoma (Caco2) cells, when grown on semipermeable filters, spontaneously differentiate in culture to form confluent monolayers which both structurally and functionally resemble the small intestinal epithelium". Do they have any evidence in literature?

Response:

We appreciate your comment regarding the characterization of Caco2 cells and their differentiation in culture. The statement that Caco2 cells, when grown on semipermeable filters, spontaneously differentiate to form confluent monolayers structurally and functionally resembling the small intestinal epithelium is well-supported in the literature. Several studies have demonstrated that Caco2 cells undergo spontaneous differentiation upon reaching confluence, forming tight junctions, microvilli, and expressing key transporters (such as P-glycoprotein, which we also observe in our study) that are characteristic of the small intestine epithelium (Sun H et al., 2008; Ismael J. H et al., 1996). We have now added these references to the Method section in the revised manuscript to substantiate this claim.

The manuscript has terrible English and the language needs drastic refinement.

Response:

We sincerely thank you for your constructive feedback on the language quality of our manuscript. To address this concern, we have carefully revised the text for grammar and clarity and engaged a professional language editing service. We believe that the revisions have significantly improved the language quality of the manuscript, and we are confident that the updated version is now much clearer and easier to understand.

Lastly, I still do not understand why *Odoribacter*. Although I agree that it shows consistent change in both mice and human samples, there are other *Bacteroides* as well. The sample sizes of the mice and the human cohorts here are too small. Unless they are able to replicate their three-way findings on GMDS and TSTA3 enzyme abundance,

Odoribacter splanchnicus and aging in other larger human cohorts or large mice cohorts, I cannot recommend this manuscript for publication in Nature Communications.

Response:

We sincerely thank you for this thoughtful comment. We acknowledge the limitation of sample size and agree that larger cohorts would further strengthen our findings. Regarding the choice of *O. splanchnicus*, we note that its association with aging and metabolic health is supported by recent large-cohort studies, including Ghosh et al. (Nat Med, 2024), which showed a consistent age-related decline linked to metabolic disturbances and disease risk. In line with this, our data demonstrate that depletion of *O. splanchnicus* correlates with impaired intestinal P-gp function and xenobiotic clearance. We have now cited this study and included its relevant figures and data below to further contextualize our findings.

Updated Text for Discussion (Page 12, Line 279~281 in the revised manuscript):

“*O. splanchnicus* is a beneficial gut commensal whose abundance declines with age and deteriorating health, a trend we observed in our study and that was independently validated in a large cohort of 10,207 individuals aged 40–93 years with follow-up data 32 ”

Supplementary Table 9. Identification of gut microbial features (richness, uniqueness indices, and species abundances) associated with host age.

	MC1			MC2			MC3			MC4			MC5			Overall		
	Spearman's rho (two-sided)	P value	BH-adjusted P	Spearman's rho (two-sided)	P value	BH-adjusted P	Spearman's rho (two-sided)	P value	BH-adjusted P	Spearman's rho (two-sided)	P value	BH-adjusted P	Spearman's rho (two-sided)	P value	BH-adjusted P	P value	BH-adjusted P	
1 Richness	0.125846269	2.93929E-05	NA	-0.01257354	0.67504629	NA	0.02822508	0.282848259	NA	0.037784243	0.367892667	NA	0.015167046	0.770318777	NA	0.038485763	0.013095983	NA
2 Uniqueness (Pielou)	0.07393792	0.014315644	NA	0.017950484	0.548966472	NA	0.07086446	0.02828177	NA	-0.007499005	0.858211845	NA	0.039916756	0.442107354	NA	0.050996575	0.00097873	NA
3 Uniqueness (Jaccard)	0.14661724	1.13985E-06	NA	0.061126558	0.041045018	NA	0.13306673	3.63598E-05	NA	0.087895909	0.035910336	NA	0.09267135	0.074620335	NA	0.111359781	6.67109E-13	NA
4 Uniqueness (Simpson)	0.042458325	0.161108973	NA	0.004304033	0.51422666	NA	0.063961592	0.05414673	NA	0.002006849	0.433267509	NA	0.029303349	0.573661039	NA	0.053600282	0.000532372	NA
5 CoproBacter_ferroglossus	-0.04113382	0.144070886	0.232818552	-0.028993493	0.332981416	0.52141276	-0.009872228	0.855960246	0.514833819	0.038146622	0.363313303	0.585978707	0.034759246	0.503328601	0.748018564	-0.015934575	0.304666186	0.38872845
6 Butyrivibrio_symbiogenus	0.002783254	0.900440021	0.956704538	-0.036056778	0.22854487	0.408549237	-0.036802038	0.255129295	0.511292784	0.076887248	0.066602893	0.180484386	0.045621614	0.379578487	0.715425272	-0.004933002	0.733641029	0.718018685
7 Butyrivibrio_viriosus	0.02386444	0.429573737	0.556739629	-0.015728151	0.959511521	0.76274886	-0.045460542	0.15538784	0.40988103	0.06904692	0.095449182	0.224620927	0.031946799	0.538475562	0.772209265	-0.00065114	0.971021938	0.580752979
8 Odoribacter_splanchnicus	-0.06610294	0.061761796	0.120789617	-0.037919179	0.569991778	0.333376436	-0.01926902	0.6863384	0.871734465	0.029819071	0.477394356	0.67947921	-0.02680027	0.444465694	0.966145944	-0.03703461	0.2004481	0.38017648
9 Paraprovetella_citris	0.063442513	0.033145873	0.080424845	0.03273108	0.274934789	0.461897895	0.049814111	0.13373046	0.371960527	0.006209592	0.882399618	0.965108819	-0.03719201	0.47429054	0.748018564	0.038728124	0.012465818	0.04846877
10 Paraprovetella_yilonghilli	0.065314391	0.030564267	0.079110927	0.018893507	0.716947886	0.837129092	0.03209101	0.19208828	0.591746633	0.013927244	0.740044566	0.905993953	-0.029077148	0.575611406	0.80232095	0.002188909	0.060226691	0.097575004
11 Prevotella_cappi	0.08210476	0.00613103	0.021429774	0.016440017	0.58078724	0.76690157	0.00386108	0.79605253	0.894312898	0.046112362	0.417620946	0.44848474	-0.01959374	0.644642495	0.846145044	0.017453021	0.076578261	0.120850037
12 Prevotella_sp_CAG_279	0.069125958	0.02210243	0.061160149	0.068080498	0.04158436	0.170320082	0.031978949	0.322780838	0.592742994	0.066383313	0.113387876	0.257352259	0.125701023	0.015132841	0.296700897	0.002552623	0.58548E-05	0.000236493

A representative excerpt of Fig.3d and Supplementary Table 9 in Ghosh et al. (2024).

Identification of MC-related and age-related gut microbial species in JD_2014. Left, heatmap for MC-related species and their associations with host variables in 2,929 drug-negative participants. Right, heatmap for age-related species and their associations with host variables in MC1 participants (partial Spearman's rank correlation analysis after adjusting for sex and environmental factors). Associations between 21 metabolic variables and microbial species were assessed using partial Spearman's rank correlation, with adjustment for age, sex and environmental factors. Asterisks denote a Benjamini–Hochberg-adjusted P < 0.05.

To address the concern regarding sample size and reproducibility in mice, we emphasize that our revised manuscript now includes validation in an expanded murine cohort (n=20 per group, young vs. old mice). As shown in Figure 2A–J, both protein and mRNA levels of intestinal P-gp (*ABCB1*) were significantly reduced in aged mice compared to young controls, and these findings were consistently supported by IHC staining. Importantly, correlation analyses further confirmed that the abundance of *O. splanchnicus* in small intestinal contents positively associates with jejunal and ileal P-gp expression, as well as with *Gmd* and *WcaG* DNA levels (Figure S7d–g). Moreover, we observed that young mice exhibited significantly higher expression of GMDS and TSTA3 in feces and intestinal contents (Figure S6d–h), both of which are critical enzymes for GDP-L-fucose biosynthesis. These findings were validated by western blotting and qPCR analysis of bacterial supernatants (Figure S7a–c), providing mechanistic evidence that *O. splanchnicus* itself expresses GMDS and TSTA3 and directly produces GDP-L-fucose. Taken together, these data demonstrate that our three-way findings (microbiome changes, GMDS/TSTA3 abundance, and P-gp expression) are robustly reproduced in larger murine cohorts, thereby strengthening the causal link between *O. splanchnicus*, GDP-L-fucose metabolism, and intestinal P-gp regulation during aging.

Figure 2A-F.

The expression of intestinal P-gp significantly declines in aged mice, closely associated with gut microbiota changes and has been validated using human intestinal cells.

A. The expression levels of P-gp in jejunum of young mice group (3-month-old mice, mus-y) and old mice group (22-26-month-old mice, mus-o) were measured by western blotting (n = 20/group).

B. The relative expression levels of proteins in jejunum were analyzed using Image J software (n = 20/group); ****p < 0.0001, two-tailed unpaired t-test.

C. Comparison of the mRNA expression levels of *Abcb1* in the jejunum of mus-y and mus-o groups. (n=20/group); *p < 0.05, Mann-Whitney U test.

D. The expression levels of P-gp in ileum of mus-y and mus-o groups were measured by western blotting (n = 20/group).

E. The relative expression levels of proteins in ileum were analyzed using Image J software (n = 20/group); ***p < 0.001, two-tailed unpaired t-test with Welch's correction.

F. Comparison of the mRNA expression levels of *Abcb1* in the ileum of mus-y and mus-o groups (n=20/group); **p < 0.01, Mann-Whitney U test.

Reviewer #2 (Remarks to the Author):

In their work, Cui et al. propose aging-related changes in the abundance of a particular bacterial species, *Odoribacter splanchnicus*, as a causative agent in aging-associated suppression of P-glycoprotein (P-gp). P-gp in turn is an important transporter involved in xenobiotic efflux in the small intestine and its reduced expression with age is associated with an increased likelihood of adverse drug reactions due to less efficient detoxification/disposal of pharmaceuticals. On the host side, they show that L-fucose is involved in a signaling cascade that induces P-gp expression in epithelial cells. I find this work elegantly addresses a quite important aspect of microbiome-host-interactions in aging. However, I do think that some of the aspects described in the work are not fully supported by the data.

Response to Reviewer #2:

We sincerely thank you for the thoughtful and constructive evaluation of our work. We are very encouraged that you recognize the novelty and importance of our study in elucidating the microbiome–host interactions underlying age-related suppression of intestinal P-glycoprotein (P-gp). We fully agree with you that certain aspects of our data presentation and interpretation required clarification and additional evidence. In the revised manuscript, we have substantially strengthened our analyses, added new validation experiments, and refined the discussion to more clearly delineate the mechanistic role of *Odoribacter splanchnicus* (*O. splanchnicus*) and GDP-L-fucose in regulating P-gp. Below, we provide detailed point-by-point responses to all comments.

Major points

1) The authors identify *O. splanchnicus* as a main producer of L-fucose in both humans and mice and argue that the reduced abundance of this species is causally involved in aging-associated decline of P-gp. However, this mechanistic connection is only poorly supported by the data. They observe both in humans as well as in mice, a large number of aging-regulated species. Particularly, they observe a suppression of *Bacteroides* species. On the other hand, they find that *Bacteroides* are a major clade containing the enzymes for L-fucose production from GDP-D-mannose. Moreover, they find the relevant enzymes also in many other species present in the gut such as *Bacillota* or *Pseudomonadota*. Even if those enzymes seem to be enriched in *Bacteroides*, this does not necessarily prove that the demise of this particular species actually mediates aging-associated decline of L-fucose production. They do show that the aging microbiome modulates P-gp expression and that L-fucose exerts an effect on P-gp expression, but the data does not support that it is specifically L-fucose production by *O. splanchnicus* that mediates the effect of the aging microbiome on P-gp expression.

Response:

We sincerely thank you for this insightful comment. We agree that enzymes involved in GDP-L-fucose biosynthesis are present in multiple gut taxa, including *Bacteroidetes*,

Bacillota, and *Pseudomonadota*. However, our revised analyses consistently highlight *O. splanchnicus* as the most relevant species in the context of aging-associated decline of intestinal P-gp. Specifically, in an expanded mouse cohort (n=20 per group), we confirmed that *O. splanchnicus* exhibited a robust and reproducible age-related depletion, whereas 10 other representative *Bacteroides* and *Parabacteroides* species showed no significant differences in abundance between young and aged mice (Fig. S15a–j). Furthermore, we analyzed human fecal metagenomic data for the same 10 species and similarly observed no significant differences between young (n = 10) and older (n=9) cohorts (Fig. S15k), further supporting the specificity of the aging-associated decline in *O. splanchnicus* rather than other *Bacteroides* species contributing to GDP-L-fucose production.

Updated Text for Discussion (Page 12, Line 273~278 in the revised manuscript):

“Importantly, our cross-species analyses identified *O. splanchnicus* as the most consistently depleted taxon with aging, whereas ten other representative *Bacteroides* and *Parabacteroides* species showed no significant differences between young and old mice or humans (Figs. S15a–k). This specificity suggests that the decline of *O. splanchnicus*, rather than general loss of *Bacteroides*, underlies the reduction in GDP-L-fucose biosynthesis and P-gp activity.”

Figure S15

Other Bacteroidetes species exhibit no differential abundance in young vs. aged mouse small intestinal contents or human feces.

a-j. The abundance of *Bacteroides_vulgatus*, *Bacteroides_stercoris*, *Bacteroides_eggerthii*, *Bacteroides_fragilis*, *Bacteroides_uniformis*, *Bacteroides_caccae*, *Bacteroides_thetaiotaomicron*, *Bacteroides_ovatus*, *Parabacteroides_merdae*, and *Parabacteroides_distasonis* in small intestine contents bacterial DNA from mus-y and mus-o groups by qPCR (n=20/group); t test (and nonparametric test).

k. The abundance of *Bacteroides_vulgatus*, *Bacteroides_stercoris*, *Bacteroides_eggerthii*, *Bacteroides_fragilis*, *Bacteroides_uniformis*, *Bacteroides_caccae*, *Bacteroides_thetaiotaomicron*, *Bacteroides_ovatus*, *Parabacteroides_merdae*, and *Parabacteroides_distasonis* in feces from HY and HO groups by metagenomics (n=9-10/group); t test (and nonparametric test).

To provide causal evidence beyond correlative analyses, we conducted supplementation experiments in naturally aged C57BL/6 mice (22–24 months). Mice were pretreated with antibiotics for 2 weeks and then gavaged with either PBS control (Abx-o-Con) or *O. splanchnicus* (Abx-o-OS) for 2 weeks (n=8/group), with successful colonization confirmed by qPCR of bacterial DNA in intestinal contents. *O. splanchnicus* gavage significantly elevated P-gp expression in both the jejunum and ileum (Fig. S5). In parallel, correlation analyses (n=20/group) showed that *O. splanchnicus* expresses the key enzymes GMDS and TSTA3, produces GDP-L-fucose as confirmed by LC-MS, and that its abundance positively correlates with P-gp (*ABCB1*) expression in both jejunum and ileum (Fig. S7). Moreover, *O. splanchnicus* abundance correlated with DNA levels of *Gmd* and *WcaG*, as well as functional orthologs (COG1089, K01711, K02377) detected in metagenomic datasets, further supporting its enzymatic role in GDP-L-fucose biosynthesis. Together, these results provide both correlative and functional validation that *O. splanchnicus* colonization is sufficient to restore intestinal P-gp expression in aged hosts. We have clarified the rationale and provided additional supporting analyses. These revisions are now incorporated into the Results section of the revised manuscript. We hope these additional experimental results address your concerns and strengthen the support for the mechanistic connection between *O. splanchnicus*, GDP-L-fucose production, and aging-associated changes in P-gp expression.

Updated Text for Results (Page 7, Line 150~163 to Page 9, Line 188~191 in the revised manuscript):

“P-gp Upregulation by *O. splanchnicus* and GDP-L-fucose

To identify specific gut microbes responsible for regulating intestinal P-gp, we profiled human fecal metagenomes (10 young adults, HY; 9 elderly, HO, Table S2) and murine 16S rRNA sequencing (10 juvenile, mus-y; 10 aged, mus-o). Both analyses revealed marked restructuring of microbial communities with age, as reflected by beta-diversity analyses at family, order, and genus levels (Figs. 3A–C, 3E–F, and S3a–d). This decline was accompanied by consistent depletion of *Odoribacter*, particularly *O. splanchnicus*, in both species (Figs. 3D–L and S3e–f). Independent validation using PCR and qPCR confirmed a significant reduction of *O. splanchnicus* in the feces and small intestinal contents of aged mice (Fig. S3g–l). Cross-species prioritization thus highlighted *O. splanchnicus* as the most robust age-depleted taxon potentially linked to intestinal P-gp

expression.

To evaluate its functional role, 3-month-old C57BL/6 mice were pretreated with antibiotics and subsequently colonized with either PBS (Abx-Control) or *O. splanchnicus* (Abx-OS) (Figs. 4A and S4a–d). Abx-OS mice exhibited significantly increased P-gp protein and *Abcb1* mRNA levels in jejunum and ileum, confirmed by Western blot, qPCR, and IHC (Figs. 4B–G and S4e–h). Similar restoration was observed in aged, antibiotic-pretreated mice gavaged with *O. splanchnicus* (Fig. S5a–f).

Untargeted metabolomic profiling of murine small intestine demonstrated that the relative abundance of GDP-L-fucose was markedly higher in young mice than in aged mice ($\log_2FC = 5.38$), whereas other microbial metabolites, including short-chain fatty acids and related derivatives, declined modestly (Fig. 4H). KEGG pathway enrichment consistently highlighted fructose/mannose metabolism as the dominant age-associated pathway across human metagenomes, murine 16S data, and intestinal metabolomics (Figs. 4I–J and S6a–b). Key enzymes in this pathway, GDP-mannose 4,6-dehydratase (GMDS, COG1089, K01711) and GDP-L-fucose synthase (TSTA3, K02377), were significantly enriched in young groups (Fig. 4K–L).

Survey of 1,647 Human Microbiome Project genomes revealed that GMDS and TSTA3 are prevalent in *Bacteroidota*, including *O. splanchnicus* (Fig. S6c). Western blot and qPCR confirmed higher GMDS/TSTA3 expression in feces and small intestine contents of young versus aged mice (Fig. S6d–h). Colonization with *O. splanchnicus* increased GMDS/TSTA3 levels (Fig. S6i–n), and both enzymes were detectable in *O. splanchnicus* itself (Fig. S7a). Colonization with *O. splanchnicus* promoted GDP-L-fucose production, verified by LC–MS (Fig. S7b–c). *O. splanchnicus* abundance correlated positively with *Gmd/WcaG* levels, supporting an *O. splanchnicus* – GMDS/TSTA3–P-gp regulatory axis (Fig. S7d–i).

Functional validation demonstrated that exogenous GDP-L-fucose (250 μ M, 48 h) significantly upregulated P-gp expression in Caco2, LS180 and T84 cells (Figs. 4M–N and S8a–d). In vivo, antibiotic-pretreated mice gavaged with GDP-L-fucose (150 mg/L, 2 weeks) exhibited increased P-gp and *Abcb1* expression in jejunum and ileum (Figs. 4O–U, S8e–h).

Overall, these findings indicate that *O. splanchnicus* regulates intestinal P-gp expression via the GMDS/TSTA3-GDP-L-fucose axis. Age-related decline in *O. splanchnicus* and the associated biosynthetic pathway likely contributes to diminished P-gp function in older adults, providing mechanistic insight into microbiota-mediated regulation of intestinal drug transport.”

2) L-fucose is also an important component of mucins that is released by mucin-degrading bacteria. Since there is also an aging-associated decline in mucin expression, can the authors exclude that the effect of fecal water from aged participants on P-gp expression is not primarily driven by reduced L-fucose release due to reduced mucin availability in the old?

Response:

We sincerely appreciate your comment regarding the potential role of mucin degradation and reduced GDP-L-fucose release in the aging-associated decline of P-gp expression. We understand that mucins, being an important source of GDP-L-fucose, may contribute to GDP-L-fucose availability in the gut, and aging-associated reductions in mucin expression could potentially influence P-gp expression. To address this concern, we conducted several experiments to examine whether the observed effects of fecal water from aged participants on P-gp expression are driven primarily by reduced GDP-L-fucose release due to decreased mucin availability.

We measured the expression of *Muc2* mRNA in small intestine (jejunum and ileum) and colon tissues from both young and aged mice. While we observed no significant differences in *Muc2* expression in the jejunum and ileum, we found a significant decline in *Muc2* expression in the colon of aged mice (see figure and table below). This suggests that the reduction in mucin expression with aging is more pronounced in the colon, consistent with prior reports. To assess whether the aging-associated decline in *MUC2* could affect P-gp expression, we performed *MUC2* knockdown experiments in Caco2 cells. After silencing *MUC2*, we observed no significant changes in P-gp (*ABCB1*) mRNA levels, indicating that reduced *MUC2* expression does not directly impact P-gp expression. These findings suggest that the decline in mucin expression in aging individuals, although notable in the colon, does not seem to directly contribute to the observed reduction in P-gp expression. Taken together, our data provide strong evidence that reduced mucin availability, particularly the decline in *MUC2* expression in the colon, is unlikely to be the primary driver of the reduction in P-gp expression in aging. We believe these findings further strengthen our hypothesis that microbial-derived GDP-L-fucose, rather than mucin degradation, plays a central role in regulating P-gp expression during aging.

Figure.
Intestinal *ABCB1* expression is unaffected by mucin.

a-c. The mRNA levels of *Muc2* in jejunum, ileum, and colon of mus-y and mus-o groups were analyzed by qPCR (n = 20/group); *p < 0.05, t test (and nonparametric test).

d. The mRNA levels of *MUC2* and *ABCB1* in Caco2 cells transfected with si-NC or si-MUC2 for 48 h (n=6/group); **p < 0.01, two-way ANOVA.

Table. siRNA and primer sequences for MUC2

Primer name	Sequence(5'to3')
h-MU2-F	TCAAAAGCAGCGTGTTTCAGC
h-MU2-R	GAGTTGGTACACACGCAGGA
SiRNA name	Sequence(5'to3')
MUC2:	
Sense strand:	GCCUCACCCUCAUGUGGAATT
Antisense strand:	UCCACAUGAGGGUGAGGCTT

3) Can the authors discuss a bit more about the origins of GDP-D-mannose that is the precursor of L-fucose. Is it produced/released by host cells or synthesized de novo by bacteria?

Response:

We thank you for raising this important point regarding the origin of GDP-D-mannose. To clarify, we systematically evaluated whether host biosynthetic capacity is affected by aging or microbial interventions. Specifically, the expression of GMDS and TSTA3 in jejunum, ileum, and colon showed no significant changes between young and aged mice, nor after fecal microbiota transplantation, *O. splanchnicus* colonization, or GDP-L-fucose supplementation (Figs. S13 and S14). These findings indicate that host-derived GDP-D-mannose biosynthesis remains intact during aging. Therefore, the aging-associated decline in GDP-L-fucose availability is unlikely to originate from the host but instead reflects reduced microbial contributions, particularly from *O. splanchnicus*.

Updated Text for Discussion (Page 13, Line 298~301 in the revised manuscript):

“Although host cells can synthesize GDP-L-fucose via GMDS and TSTA3, our data show that expression of these enzymes remains stable across age groups and is unaffected by microbiota interventions (Figs. S13–14)⁴³. This indicates that age-related decline in GDP-L-fucose availability primarily arises from microbial, rather than host, factors.”

Figure S13

Intestinal GMDS and TSTA3 expression levels are unaffected by age or fecal microbiota transplantation in mice.

a-b. The expression levels and quantitative analysis of GMDS and TSTA3 in jejunum of mus-y and mus-o groups (n = 20/group); two-way ANOVA.

c. The mRNA levels of *Gmd* and *Tsta3* in jejunum of mus-y and mus-o groups were analyzed by qPCR (n = 20/group); two-way ANOVA.

d-e. The expression levels and quantitative analysis of GMDS and TSTA3 in ileum of mus-y and mus-o groups (n = 20/group); two-way ANOVA.

f. The mRNA levels of *Gmd* and *Tsta3* in ileum of mus-y and mus-o groups were analyzed by qPCR (n = 20/group); two-way ANOVA.

g-h. The expression levels and quantitative analysis of GMDS and TSTA3 in colon of mus-y and mus-o groups (n = 20/group); two-way ANOVA.

i. The mRNA levels of *Gmd* and *Tsta3* in colon of mus-y and mus-o groups were analyzed by qPCR (n = 20/group); two-way ANOVA.

j-k. The expression levels and quantitative analysis of GMDS and TSTA3 in jejunum of Abx-mus-yy and Abx-mus-oy groups (n = 8/group); two-way ANOVA.

l-m. The expression levels and quantitative analysis of GMDS and TSTA3 in ileum of Abx-mus-yy and Abx-mus-oy groups (n = 8/group); two-way ANOVA.

n-o. The expression levels and quantitative analysis of GMDS and TSTA3 in colon of Abx-mus-yy and Abx-mus-oy groups (n = 8/group); two-way ANOVA.

For logical continuity and rigor, data originate from the same experiment and sample set as prior related studies.

Figure S14
Intestinal GMDS and TSTA3 expression levels are unaffected by *O. splanchnicus* or GDP-L-fucose in mice.

a-b. The expression levels and quantitative analysis of GMDS and TSTA3 in jejunum of Abx-Control and Abx-OS groups (n = 7-10/group); two-way ANOVA.
c-d. The expression levels and quantitative analysis of GMDS and TSTA3 in ileum of Abx-Control and Abx-OS groups (n = 7-10/group); two-way ANOVA.
e-f. The expression levels and quantitative analysis of GMDS and TSTA3 in colon of Abx-Control and Abx-OS groups (n = 7-10/group); two-way ANOVA.
g-h. The expression levels and quantitative analysis of GMDS and TSTA3 in jejunum of Abx-o-Con and Abx-o-OS groups (n = 8/group); two-way ANOVA.
i-j. The expression levels and quantitative analysis of GMDS and TSTA3 in ileum of Abx-o-Con and Abx-o-OS groups (n = 8/group); two-way ANOVA.
k-l. The expression levels and quantitative analysis of GMDS and TSTA3 in colon of Abx-o-Con and Abx-o-OS groups (n = 8/group); two-way ANOVA.
m-n. The expression levels and quantitative analysis of GMDS and TSTA3 in jejunum of Control and GDP-L-fucose groups (n = 10/group); two-way ANOVA.
o-p. The expression levels and quantitative analysis of GMDS and TSTA3 in ileum of Control and GDP-L-fucose groups (n = 10/group); two-way ANOVA.
q-r. The expression levels and quantitative analysis of GMDS and TSTA3 in colon of Control and GDP-L-fucose groups (n = 10/group); two-way ANOVA.
For logical continuity and rigor, data originate from the same experiment and sample set as prior related studies.

4) The authors state that the major site of action of L-fucose mediated suppression of P-gp is the small intestine. However, microbial abundance in this part of the gastrointestinal tract is way lower than in the colon. It should be discussed which role colonic expression of P-gp plays in the observed increased likelihood of adverse drug reactions due to P-gp suppression in age.

Response:

We thank you for this insightful comment regarding the relative contributions of small intestinal versus colonic P-gp in aging. We agree that microbial density is substantially higher in the colon, and therefore it is important to clarify whether colonic P-gp also plays a role. In our revised manuscript, we now provide new experimental data (Fig. S12) showing that colonic P-gp expression is indeed reduced in aged mice compared to young controls (Fig. S12a–c). Importantly, colonic P-gp expression could be restored by fecal microbiota transplantation from young donors, *O. splanchnicus* supplementation, GDP-L-fucose gavage, or EGW treatment (Fig. S12d–q). These findings indicate that the regulatory effect of *O. splanchnicus* and GDP-L-fucose extends to the colon as well.

Updated Text for Discussion (Page 11, Line 246~252 in the revised manuscript):

“We also observed that colonic P-gp expression declines with aging and can be restored by young microbiota, *O. splanchnicus*, GDP-L-fucose, or engineered bacteria (Fig. S12). Importantly, the regulatory patterns observed in the colon are consistent with those in the small intestine, where P-gp also declines with age. From a pharmacological perspective, small intestinal P-gp plays a predominant role in systemic drug absorption and the risk of adverse drug reactions, underscoring its critical importance in maintaining drug homeostasis²².”

Figure S12

Young mouse-derived fecal microbiota transplantation, *O. splanchnicus*, GDP-L-fucose, or EGW promote colonic P-gp expression in mice.

a-b. The expression levels and quantitative analysis of P-gp in colon of mus-y and mus-o groups (n = 20/group); **p < 0.01, t test (and nonparametric test).

c. The mRNA levels of *Abcb1* in colon of mus-y and mus-o groups were analyzed by qPCR (n = 20/group); *p < 0.05, t test (and nonparametric test).

d-e. The expression levels and quantitative analysis of P-gp in colon of mus-yy and mus-oy groups (n = 6/group); **p < 0.01, t test (and nonparametric test).
f. The mRNA levels of *Abcb1* in colon of mus-yy and mus-oy groups were analyzed by qPCR (n = 6/group); **p < 0.01, t test (and nonparametric test).
g-h. The expression levels and quantitative analysis of P-gp in colon of Abx-mus-yy and Abx-mus-oy groups (n = 8/group); ***p < 0.001, t test (and nonparametric test).
i-j. The expression levels and quantitative analysis of P-gp in colon of Abx-Control and Abx-OS groups (n = 7-10/group); **p < 0.01, t test (and nonparametric test).
k-l. The expression levels and quantitative analysis of P-gp in colon of Abx-o-Con and Abx-o-OS groups (n = 8/group); **p < 0.01, t test (and nonparametric test).
m-n. The expression levels and quantitative analysis of P-gp in colon of Control and GDP-L-fucose groups (n = 10/group); **p < 0.01, t test (and nonparametric test).
o-p. The expression levels and quantitative analysis of P-gp in colon of Abx-Control, Abx-EC, and Abx-EGW groups (n = 6/group); *p < 0.05, one-way ANOVA.
q. The mRNA levels of *Abcb1* in colon of Abx-Control, Abx-EC, and Abx-EGW groups were analyzed by qPCR (n = 6/group); *p < 0.05, **p < 0.01, one-way ANOVA.

5) Also along these lines, most of the microbiome data the authors generate are from fecal samples and thus from colon. Since microbiota from colon and small intestine differ and *Bacteroides* levels in small intestine are typically smaller than in the colon, can they actually show that *O. splanchnicus* is actually present in the small intestine and declining with age? I guess corresponding data for humans might be hard to come by, but at least in mice this should be possible to determine.

Response:

Thank you for this thoughtful comment. We agree that distinguishing between colonic and small intestinal microbiota is important, as their compositions differ substantially. In response, we extended our analyses to include small intestinal contents in mice. As described in the revised Results and shown in Fig. S3g–S3l, independent validation using PCR and qPCR confirmed that *O. splanchnicus* is indeed present in the small intestine and undergoes age-associated depletion, consistent with fecal findings. Moreover, correlation analyses demonstrated that the abundance of *O. splanchnicus* in small intestinal contents positively correlates with P-gp expression in both young and aged mice (Fig. S7d–e).

In addition, untargeted metabolomics of small intestinal tissue revealed GDP-L-fucose as the most significantly decreased metabolite in aged mice, with pathway enrichment analyses highlighting fructose/mannose metabolism as the dominant age-associated pathway across datasets (Figs. 4H–J, S6a–b). Survey of Human Microbiome Project genomes further showed that GMDS and TSTA3 are enriched in *Bacteroidota*, including *O. splanchnicus* (Fig. S6c), and experimental validation confirmed higher *Gmd/WcaG* levels in fecal and small intestinal contents of young mice (Fig. S6d–h). Colonization with *O. splanchnicus* increased GMDS/TSTA3 levels and promoted GDP-L-fucose production (Figs. S6i–n, S7b–c).

Together, these additional data address your concern and provide comprehensive evidence that *O. splanchnicus* is present in the small intestine, declines with age, and functionally contributes to intestinal P-gp regulation through the GMDS/TSTA3–GDP-L-fucose axis.

Figure S3

The gut microbiota diversity in the elderly is markedly distinct from that in younger individuals.

a-d. The Metagenomics gene profiling data for fecal microbiome from HY and HO groups (n=9-10/group).

a. PCoA plot of beta-diversity at family level.

b. Relative abundance of significantly altered taxa at the rank of family (including unspecified taxa).

c. PCoA plot of beta-diversity at genus level.

d. Relative abundance of significantly altered taxa at the rank of genus (including unspecified taxa).

e-f. The 16S rRNA gene profiling data for fecal microbiome from mus-y and mus-o groups (n=10/group).

e. Relative abundance of significantly altered taxa at the rank of genus (including unspecified taxa).

f. LEfSe analysis of significantly altered taxa at the rank of genus (including unspecified taxa).

g. The abundance of *Odoribacter_splanchnicus* (OS) in fecal bacteria DNA from mus-y and mus-o groups by PCR (n=6/group).

h. The abundance of *O. splanchnicus* in fecal bacteria DNA from mus-y and mus-o groups by qPCR (n=6/group); **p < 0.01, t test (and nonparametric test).

i. The abundance of *O. splanchnicus* in fecal bacteria DNA from HY and HO groups by PCR (n=23/group).

j. The abundance of *O. splanchnicus* in fecal bacteria DNA from HY and HO groups by qPCR(n=23/group); **p < 0.01, t test (and nonparametric test).

k. The abundance of *O. splanchnicus* in small intestine contents bacterial DNA from mus-y and mus-o groups by PCR (n=20/group).

l. The abundance of *O. splanchnicus* in small intestine contents bacterial DNA from mus-y and mus-o groups by qPCR (n=20/group); ****p < 0.0001, t test (and nonparametric test).

a**b****c****d****e****f****g****h****i****j****k****l****m** Small intestine contents**n** Small intestine contents
Figure S6

Young mice's feces and small intestine contents exhibit high expression of GMDS and TSTA3.

- a. The KEGG level3 LEfSe analysis of metagenomic genes for fecal microbiome from HY and HO groups (n=9-10/group).
- b. The KEGG level3 LEfSe analysis of 16S rRNA gene profiling data for fecal microbiome from mus-y and mus-o groups (n=10/group).
- c. Distribution of GMDS and TSTA3 in Human Microbiome Project (HMP) reference genomes. The pie charts show the total number of microbial genomes harboring the corresponding subject (GMDS and TSTA3) classified according to the phyla. The leftmost chart shows the total number of microbial genomes included in the analyses for each phylum. Analyses were performed using the COG and KEGG functions; COG1089 (for GMDS) and K02377 (for TSTA3), while K01711 (for GMDS) was discarded due to low data accuracy.
- d. The expression levels of GMDS and TSTA3 in feces of mus-y and mus-o groups. (n=6/group)
- e. The relative expression levels of GMDS in feces were analyzed using Image J software (n=6/group); ****p < 0.0001, t test (and nonparametric test).
- f. The relative expression levels of TSTA3 in feces were analyzed using Image J software (n=6/group); **p < 0.01, t test (and nonparametric test).
- g. Detection of *Gmd* levels in bacterial DNA from small intestinal contents of mus-y and mus-o groups by qPCR (n=20/group); ****p < 0.0001, t test (and nonparametric test).
- h. Detection of *WcaG* levels in bacterial DNA from small intestinal contents of mus-y and mus-o groups by qPCR (n=20/group); ***p < 0.001, t test (and nonparametric test).
- i. The expression levels of GMDS and TSTA3 in feces of Abx-Control and Abx-OS groups (n=6/group).
- j. The relative expression levels of proteins in feces were analyzed using Image J software (n=6/group); ***p < 0.001, ****p < 0.0001, two-way ANOVA.
- k. Detection of *Gmd* levels in bacterial DNA from small intestinal contents of Abx-Control and Abx-OS groups by qPCR (n=7-10/group); ***p < 0.001, t test (and nonparametric test).
- l. Detection of *WcaG* levels in bacterial DNA from small intestinal contents of Abx-Control and Abx-OS groups by qPCR (n=7-10/group); ***p < 0.001, t test (and nonparametric test).
- m. Detection of *Gmd* levels in bacterial DNA from small intestinal contents of Abx-o-Con and Abx-o-OS groups by qPCR (n=8/group); **p < 0.01, t test (and nonparametric test).
- n. Detection of *WcaG* levels in bacterial DNA from small intestinal contents of Abx-o-Con and Abx-o-OS groups by qPCR (n=8/group); ***p < 0.001, t test (and nonparametric test).

Minor:

- Fig. 1E - doesn't really contain information and can be removed
- Fig. 1F-I: font sizes in the figure are too small to be readable, particularly in Fig. 1I
- Fig. 3H and I not readable

Response:

We sincerely appreciate your valuable feedback on the figures. We have made the following revisions based on your suggestions:

Fig. 1E: We agree with you that Fig. 1E does not contribute substantial information to the manuscript. Therefore, we have removed this figure in the revised version of the manuscript to streamline the presentation and enhance clarity.

Fig. 1F-I: We have increased the font size in Fig. 1F-I, particularly in Fig. 1I, where the text was previously too small to be clearly readable. The revised figure now has larger, more legible font sizes to ensure clarity and ease of reading.

Fig. 3H and I: We acknowledge that Figs. 3H and 3I were not sufficiently clear in their previous form. We have improved the resolution to ensure that the figures are now readable and easier to interpret.

We believe these revisions have significantly improved the quality and clarity of the figures, and we hope the changes meet your expectations.

Reviewer #3 (Remarks to the Author):

Chen Cui et al report on *Doribacter splanchnicus* which rescues aging-related intestinal P2 glycoprotein damage via GDP-L-Fucose secretion. Specifically, they propose gut microbiota dysbiosis as a key driver of age-associated P-gp deficiency. They performed multi-omics analyses of human cohorts and murine models. They find that *Odoribacter splanchnicus* (*O. splanchnicus*) has a superior enzymatic activity and capacity to enzymatically provide the relevant biosynthesis of the microbial derived metabolite GDP-L-Fucose. This in turn should then result in GDP-L-Fucose activating c-Jun-mediated transcriptional programming through phosphorylation-dependent enhancement of eukaryotic translation initiation factor 4E (eIF4E), thereby upregulating ABCB1 expression and - most importantly, restoring P-gp-mediated xenobiotic clearance. Aged humans and senescent mice display parallel depletion of colonic *O. splanchnicus* and GDP-L-Fucose levels appeared to display parallel depletion of colonic *O. splanchnicus* and GDP-L-Fucose levels that are considered as causative factors in P-gp dysfunction. The authors therefore postulate a microbiota-metabolite signaling axis (*O.splanchnicus*→GDP-L-Fucose→eIF4E/c-Jun→ABCB1). As a consequence, the authors propose and hope to prevent adverse drug reactions in the elderly by microbiome targeted interventions.

The question is important and microbiome targeted interventions are timely, particularly in the elderly with a changing composition of the gut microbiome and the risk of ADR. A potential reduction of ADR by "correction" of the gut microbiome appears therefore attractive. The experiments appear to have been performed carefully and are described in detail. Mechanistic approaches by specific transplants and ex vivo as well as in vivo data are logically provided.

Response to Reviewer #3

We sincerely thank you for the thoughtful and encouraging evaluation of our work. We greatly appreciate the recognition of the importance of microbiota-targeted interventions in aging and the positive assessment of our mechanistic and experimental approaches. We agree that our findings highlight a timely and clinically relevant microbiota-metabolite-host signaling axis with potential implications for preventing adverse drug reactions in the elderly. In the revised manuscript, we have carefully addressed your specific concerns point by point and incorporated additional experimental evidence, clarifications, and expanded discussion to strengthen our conclusions.

The following points should be considered:

1. Intro: The role of P-GP is overfocussed and overemphasized in the elderly, since eg for DOACs many other mechanisms contribute to different plasma concentrations including liver insufficiency, cardiac function, altered kidney function polymedication and medication interactions etc etc. This is later enumerated and mentioned that it was corrected for in earlier work, but this correction may be not trivial given the continuum of organ dysfunction.

2. Particularly bleeding complications should be attributed carefully, given the many concomitant aspects involved (including eg dual anticoagulation and other medication interactions.eg line 127.
3. Likewise, elderly people have per se higher bleeding risks, even in the absence of concomitant P-GP interfering drugs.

Response:

We thank you for raising these important points. We fully agree that adverse drug reactions (particularly bleeding events with DOACs) in the elderly are multifactorial and cannot be attributed to P-gp decline alone. Indeed, organ dysfunction (renal, hepatic, cardiac), polypharmacy, and comorbidities contribute substantially to altered pharmacokinetics and bleeding risk. In our revised Introduction, we now explicitly acknowledge this broader context. At the same time, our analyses across multiple datasets consistently indicate that reduced P-gp activity represents one important mechanistic factor that exacerbates bleeding risk in the elderly, particularly when combined with other vulnerabilities. Thus, we position P-gp decline not as the sole explanation, but as a mechanistically defined contributor to age-associated susceptibility to adverse drug reactions.

Updated Text for Introduction (Page 4, Line 68–87 in the revised manuscript):

“Physiological aging is accompanied by a progressive decline in xenobiotic clearance pathways, including impaired hepatic metabolism and renal excretion, which contribute to altered pharmacokinetics and heightened susceptibility to adverse drug reactions (ADRs) in older adults ¹. Among the various age-related changes, intestinal P-glycoprotein (P-gp/*ABCB1*), an ATP-driven efflux transporter located at the apical membrane of enterocytes, plays a particularly important role in regulating drug absorption and bioavailability ²⁻⁴. Reduced P-gp activity increases systemic exposure of substrate drugs, thereby elevating the risk of clinically significant adverse events ⁵⁻⁷. Thus, while not the sole determinant, P-gp impairment represents a distinct and potentially modifiable mechanism contributing to age-related ADRs.

This is exemplified by direct oral anticoagulants (DOACs), where reduced intestinal P-gp activity increases bioavailability and plasma concentrations, correlating with higher rates of major bleeding. Clinically, patients aged ≥ 75 years exhibit a 1.4-fold higher incidence of major bleeding compared to younger individuals, rising to 2.0-fold for intracranial hemorrhage in octogenarians ^{7,8}. These findings demonstrate how impaired P-gp function critically shapes drug toxicity, even while acting in concert with other aging-related factors such as hepatic and renal dysfunction ⁹. Similarly, reduced P-gp activity elevates systemic exposure to digoxin, contributing to excess neurocardiac adverse events ¹⁰⁻¹². Together, these observations position impaired intestinal P-gp function as a pivotal mechanism linking altered drug exposure to ADRs in older adults, highlighting opportunities for targeted interventions to improve pharmacotherapy safety in geriatric populations.”

4. The n number in fig 1 seems very small.

Response:

We thank you for the comment. Fig. 1 is not a meta-analysis but a descriptive summary of published clinical pharmacokinetic studies on FDA-defined P-gp substrates. The apparent small “n” reflects the limited sample sizes reported in the original studies rather than our own experimental design. Given the scarcity and heterogeneity of available data, we refrained from formal effect size estimation and instead presented pooled descriptive comparisons to highlight the consistent trend of higher drug exposure in older adults. We have clarified this point in the Methods.

Updated Text for Methods (Page 15, Line 318~330 in the revised manuscript):

“Systematic review of pharmacokinetic data for P-gp substrates

We searched PubMed (up to August 2024) for clinical studies reporting pharmacokinetics of FDA-defined P-glycoprotein (P-gp) substrates (dabigatran etexilate, digoxin, edoxaban, fexofenadine) in young and older adults. Keywords were predefined, and studies were eligible if they stratified pharmacokinetic outcomes (AUC, C_{max}) by age and reported sample size, demographics, dosing, and variability. In total, we identified 7 studies on digoxin, 3 on fexofenadine, 6 on dabigatran, and 2 on edoxaban. Owing to the limited number and heterogeneity of such studies, no meta-analysis was performed. Instead, parameters were descriptively summarized and compared across age groups using GraphPad Prism v8.0. The “n” shown in Fig. 1A-B directly reflects the sample sizes of the original studies. Across all available reports, older adults consistently exhibited higher systemic exposure to P-gp substrates, supporting the hypothesis that age-related reductions in P-gp function contribute to altered drug disposition.”

5. The take home fig is oversimplified and needs revision

Response:

We thank you for this helpful comment. We agree that the current “take-home figure” may appear oversimplified and could benefit from a more comprehensive representation of our findings. In the revised manuscript, we have updated the schematic figure to better reflect the age-associated decline in *O. splanchnicus* abundance. The enzymatic capacity (GMDS/TSTA3) of *O. splanchnicus* in generating GDP-L-fucose. The downstream molecular signaling pathway (GDP-L-fucose → eIF4E phosphorylation → c-Jun activation → *ABCB1* transcription). The observed restoration of P-gp expression and xenobiotic clearance in aged mice upon *O. splanchnicus* supplementation.

We believe that the revised schematic provides a clearer and more balanced summary of the microbiota–metabolite–host axis identified in our study, while acknowledging the complexity of aging-related pharmacokinetic alterations.

6. Explain a potential future targeted intervention in the elderly either by bacterial transplantation with or without antibiotic treatment and or by providing L-Fucose.

Response:

We thank you for this constructive suggestion. We fully agree that discussing potential future targeted interventions is highly relevant and adds translational value to our findings.

Updated Text for Discussion (Page 13, Line 302~307 in the revised manuscript):

“From a translational perspective, our findings highlight two complementary strategies to enhance intestinal P-gp activity. One approach is to modulate the gut microbiota, for example through *O. splanchnicus* probiotics or fecal microbiota transplantation. Another strategy is to directly supplement the key metabolite, GDP-L-fucose, or its analogs. These interventions can be tailored to individual patient factors, including comorbidities, polypharmacy, and microbiome composition, enabling precision geriatric therapeutics^{41, 44-45}.”

7. Enthusiasm about the application of odoribacter splanchnicus must be tempered since it has been shown to associate with several serious diseases, incl intracranial, cardiovasc, kidney - please consider cite and discuss, eg Janhong Li in recent Microorganisms 2025, 13(4), 815; <https://doi.org/10.3390/microorganisms13040815>.

Response:

We thank you for this important comment and fully agree that the enthusiasm regarding *O. splanchnicus* as a therapeutic candidate should be tempered. Indeed, recent studies

have reported associations between *O. splanchnicus* and pathological conditions, including intracranial, cardiovascular, and kidney diseases (Li et al., Microorganisms, 2025). We have now incorporated this literature into the Discussion and emphasized that while *O. splanchnicus* exhibits protective effects in our study by restoring P-gp function, its role is likely context-dependent and may vary with host physiology and comorbidities.

Updated Text for Discussion (Page 13, Line 290~295 in the revised manuscript):

“Importantly, while *O. splanchnicus* is largely beneficial, isolated reports have linked it to pathological conditions, including cardiovascular, renal, and intracranial diseases, likely reflecting opportunistic behavior under severe inflammation or compromised intestinal barriers⁴¹. These findings underscore the need for careful safety evaluation in therapeutic applications, but overall support its role as a key protective commensal whose age-related depletion may impair intestinal homeostasis and drug detoxification.”

8. Under specific pathological conditions, such as exacerbated intra-abdominal inflammation or impaired intestinal barrier function, it may act as an opportunistic pathogen, triggering infections or even sepsis. Please discuss.

Response:

We thank you for raising this important point. We agree that under pathological conditions such as exacerbated intra-abdominal inflammation or impaired intestinal barrier function, *O. splanchnicus* may act as an opportunistic pathogen and potentially trigger systemic infections, including sepsis. To address this concern, we have revised the Discussion to emphasize this possibility and to highlight that any therapeutic application of *O. splanchnicus* should be preceded by rigorous safety assessments and stratification of patient populations.

Updated Text for Discussion (Page 13, Line 290~295 in the revised manuscript):

“Importantly, while *O. splanchnicus* is largely beneficial, isolated reports have linked it to pathological conditions, including cardiovascular, renal, and intracranial diseases, likely reflecting opportunistic behavior under severe inflammation or compromised intestinal barriers⁴¹. These findings underscore the need for careful safety evaluation in therapeutic applications, but overall support its role as a key protective commensal whose age-related depletion may impair intestinal homeostasis and drug detoxification.”

9. Specificity: Another mechanism of protection has been described of *O. splanchnicus* by inhibition of other pathogens like Salmonella, as reported in doi: <https://doi.org/10.1101/2024.08.23.609322>, via its secreted bacteriocin.

Response:

We thank you for pointing out this important additional mechanism. Indeed, recent work (doi: <https://doi.org/10.1101/2024.08.23.609322>) reported that *O. splanchnicus* secretes a bacteriocin that directly inhibits the growth of pathogenic bacteria such as Salmonella, highlighting its broader role in maintaining intestinal homeostasis. While our present study focuses on the metabolite-mediated regulation of P-gp via GDP-L-

fructose, these complementary findings underscore that *O. splanchnicus* may exert protective effects through multiple, non-mutually exclusive mechanisms. We have added a discussion of this aspect in the revised manuscript.

Updated Text for Discussion (Page 13, Line 286~289 in the revised manuscript):

“The protective functions of *O. splanchnicus* are multifaceted. Beyond metabolite-mediated regulation of host xenobiotic transporters, it can directly suppress enteric pathogens through bacteriocin secretion, such as against *Salmonella*, highlighting its contribution to gut homeostasis⁴⁰.”

10. Short chain fatty acids are produced by certain gut bacteria, also by *O. splanchnicus*? Were they also restoring the P-GP activity?

Response:

We thank you for this thoughtful question. Indeed, gut bacteria is capable of producing short-chain fatty acids (SCFAs). In our untargeted metabolomics of murine small intestinal tissues, we observed that SCFAs and related derivatives displayed a downward trend during aging. However, the magnitude of change was modest (maximum $\log_2FC \approx 2.2$), whereas GDP-L-fructose showed the most pronounced increase in young group ($\log_2FC = 5.38$). Importantly, metagenomic and 16S rRNA gene pathway enrichment analyses did not reveal significant alterations in SCFA-related biosynthetic pathways, while the GDP-L-fructose biosynthetic pathway was consistently downregulated. These multi-omics data demonstrate GDP-L-fructose as the dominant metabolite linking age-associated microbiota changes to intestinal P-gp decline.

Updated Text for Results (Page 8, Line 167~170 in the revised manuscript):

“Untargeted metabolomic profiling of murine small intestine demonstrated that the relative abundance of GDP-L-fructose was markedly higher in young mice than in aged mice ($\log_2FC = 5.38$), whereas other microbial metabolites, including short-chain fatty acids and related derivatives, declined modestly (Fig. 4H).”

References

Wang T, Shi Z, Ren H, et al. 2024. Divergent age-associated and metabolism-associated gut microbiomesignatures modulate cardiovascular disease risk. *Nat Med.* 30(6):1722-1731.

Ghosh TS, Shanahan F, O'Toole PW. 2022. The gut microbiome as a modulator of healthy ageing. *Nat Rev Gastroenterol Hepatol.* 19 (9):565-584.

Ghosh TS, Shanahan F, O'Toole PW. 2022. Toward an improved definition of a healthy microbiome for healthy aging. *Nat Aging.* 2(11):1054-1069.

Sato Y, Atarashi K, Plichta DR, et al. 2021. Novel bile acid biosynthetic pathways are enriched in the microbiome of centenarians. *Nature.* 599 (7885):458-464.

Dipasree Hajra, Debapriya Mukherjee, Rhea Vij, et al. 2024. *Odoribacter splanchnicus* mitigates *Salmonella*-induced gut inflammation and its associated pathogenesis via its secreted bacteriocin. *bioRxiv.* 2024.08.23.609322.

Li J, Xu J, Guo X, et al. 2025. *Odoribacter splanchnicus*-A Next-Generation Probiotic Candidate. *Microorganisms.* 13(4):815.

Sun H, Chow EC, Liu S, Du Y, Pang KS. 2008. The Caco-2 cell monolayer: usefulness and limitations. *Expert Opin Drug Metab Toxicol.* 4(4):395-411.

Ismael J. H, Li Jb. 1996. Carrier-mediated transport and efflux mechanisms in Caco-2 cells. *Advanced drug delivery reviews.* 22(1996): 53-66.

REVIEWER COMMENTS

Reviewer #1 (Remarks to the Author):

While I appreciate the efforts of the authors in refining the version 1 of this manuscript, some of my previous concerns remain.

I believe that *O.S* is associated with a reproducible age-related decline, but I am bit skeptical about the justifications provided by the authors regarding the other functional links. GMDS and TSTA3 are encoded by other members of the *Bacteroides* phylum, many of which show the trend in both human and mice (decline in old). The authors show that supplementation by O.S increases expression of these enzymes, but what about other *Bacteroides* members. Does supplementation by other *Bacteroides* members not show these trends? There are multiple such members which show decline in age? What about *Alistipes*, *Parabacteroides*, *Barnesiella*, etc? The authors need to clearly show that supplementation by any of these other members does not show these changes.

Response:

We appreciate this insightful comment. Indeed, comparative genomic analysis indicates that GMDS and TSTA3 are broadly distributed among *Bacteroidota*. Many of these taxa also show age-related decline in abundance. In our study, we selected *O. splanchnicus* as a representative commensal because it exhibited one of the strongest and most consistent age-associated reductions in both human and mouse cohorts, and its colonization robustly increased GMDS/TSTA3 expression and GDP-L-fucose levels.

We have now added a detailed clarification in the revised manuscript (Results, page 8) acknowledging that other *Bacteroides*-related species may share similar pathways.

Updated Text for Results (Page 8, Line 189-192 in the revised manuscript):

“Notably, GMDS and TSTA3 homologs were detected across multiple *Bacteroides* species within the *Bacteroidota* phylum, suggesting that this metabolic capacity may be shared by related commensals, although *O. splanchnicus* displayed one of the strongest age-associated declines and functional effects in our dataset.”

I want to add a minor correction to the responses of the authors. The results shown by the authors in page 9 of their responses pertains to the paper by Wang et al in Nature Medicine (<https://www.nature.com/articles/s41591-024-03038-y>) not Ghosh et al (which is a different study in Nature Aging also highlighting O.S). However, my response to the clarifications/observations of the authors in this specific context is this. The Supplementary table highlighted by the authors also contains the multiple Bacteroidetes phylum members, some of which show even stronger negative associations with age (especially in the MC1 or normal metabolic cluster). Hence, why *Odoribacter*?

To summarize, I am not saying that the authors' O.S (*Odoribacter splanchnicus*) is erroneous. I just want the authors to highlight that there are other members within the same clade as well who show this property (or if not provide validation experimental results to show that it is not so).

And provide a much comprehensive data-driven investigation showing the presence of GMDS and TSTA3 homologs across all clades of human gut microbes and show that O.S and its relatives are the only ones that harbor these.

Response:

We thank the reviewer for this careful observation and sincerely apologize for the miscitation in our previous response letter. We have rechecked the main text of the manuscript and confirm that the in-text reference was cited correctly.

As the reviewer notes, multiple *Bacteroides* species in the dataset from Wang et al. (2024) exhibit an age-related decline. In our study, we chose *O. splanchnicus* because it displayed one of the most consistent and statistically significant age-associated reductions across human and mouse cohorts, could be readily cultured and stably colonized in antibiotic-treated mice, and demonstrated a direct causal link to host GMDS/TSTA3 expression, GDP-L-fucose production, and P-gp regulation.

We have clarified these points and added discussion acknowledging that other *Bacteroides* taxa may share similar pathways and emphasizing the need for future comparative validation studies (Discussion, page 14).

Updated Text for Discussion (Page 14, Line 317-330 in the revised manuscript):

“Although *O. splanchnicus* exhibited a reproducible age-related decline and a clear capacity to restore intestinal P-gp expression, we acknowledge that GMDS and TSTA3 are not unique to this species. Comparative genomic analysis revealed that homologs of these enzymes are broadly distributed among members of the *Bacteroidota* phylum. In our study, *O. splanchnicus* was selected as a representative commensal with a strong and consistent negative correlation with age, and whose colonization robustly increased GMDS/TSTA3 expression and GDP-L-fucose production. While our data demonstrate that *O. splanchnicus* can rescue P-gp dysfunction through the GMDS/TSTA3-GDP-L-fucose axis, we do not exclude the possibility that other *Bacteroides*-related taxa may exert similar effects. Further comparative colonization or co-culture experiments using additional *Bacteroides* strains will be required to systematically determine the extent to which this mechanism is conserved across the phylum.

Together, these findings suggest that age-related depletion of multiple *Bacteroides* species may contribute to diminished intestinal fucosylation and P-gp function, with *O. splanchnicus* representing one key mediator within this broader microbial network.”

Reviewer #2 (Remarks to the Author):

I thank the authors for thoroughly addressing all of my concerns.

Response:

We sincerely thank the reviewer for the positive assessment and constructive feedback during the previous round of review. We are pleased that our revisions have fully addressed all concerns. We greatly appreciate the reviewer's support for the publication of our work in *Nature Communications*.

Reviewer #3 (Remarks to the Author):

The authors have responded adequately to most of the Reviewers questions. Where some doubts of the magnitude of the effects and their specificity remain, the authors have now stated this in the ms in the discussion and where appropriate. The authors have performed additional experiments and literature review to substantiate their findings and to respond to the Reviewers points.

Response:

We thank the reviewer for the constructive evaluation and for recognizing our additional experiments and literature analyses. We appreciate the reviewer's note regarding the magnitude and specificity of the observed effects. In the revised manuscript, we have further clarified these aspects in the Results and Discussion (page 8 and 14), emphasizing that while *O. splanchnicus* shows a reproducible and significant effect on intestinal P-gp regulation, other related *Bacteroides* species may share partial functional similarities.

We are grateful for the reviewer's thoughtful comments, which helped us strengthen the rigor and interpretive balance of the manuscript.

Updated Text for Results (Page 8, Line 189-192 in the revised manuscript):

“Notably, GMDS and TSTA3 homologs were detected across multiple *Bacteroides* species within the *Bacteroidota* phylum, suggesting that this metabolic capacity may be shared by related commensals, although *O. splanchnicus* displayed one of the strongest age-associated declines and functional effects in our dataset.”

Updated Text for Discussion (Page 14, Line 317-330 in the revised manuscript):

“Although *O. splanchnicus* exhibited a reproducible age-related decline and a clear capacity to restore intestinal P-gp expression, we acknowledge that GMDS and TSTA3 are not unique to this species. Comparative genomic analysis revealed that homologs of these enzymes are broadly distributed among members of the *Bacteroidota* phylum. In our study, *O. splanchnicus* was selected as a representative commensal with a strong and consistent negative correlation with age, and whose colonization robustly increased GMDS/TSTA3 expression and GDP-L-fucose production. While our data demonstrate that *O. splanchnicus* can rescue P-gp dysfunction through the GMDS/TSTA3-GDP-L-fucose axis, we do not exclude the possibility that other *Bacteroides*-related taxa may exert similar effects. Further comparative colonization or co-culture experiments using additional *Bacteroides* strains will be required to systematically determine the extent to which this mechanism is conserved across the phylum.

Together, these findings suggest that age-related depletion of multiple *Bacteroides* species may contribute to diminished intestinal fucosylation and P-gp function, with *O. splanchnicus* representing one key mediator within this broader microbial network.”